# Magnetic Field Dynamic Strategies for the Improved Control of the Angiogenic Effect of Mesenchymal Stromal Cells

**DOI:** 10.3390/polym13111883

**Published:** 2021-06-06

**Authors:** Ana C. Manjua, Joaquim M. S. Cabral, Frederico Castelo Ferreira, Carla A. M. Portugal

**Affiliations:** 1LAQV-REQUIMTE, Department of Chemistry, NOVA School of Science and Technology, Universidade Nova de Lisboa, Campus da Caparica, 2829-516 Caparica, Portugal; carina.manjua@tecnico.ulisboa.pt; 2Department of Bioengineering and iBB—Institute for Bioengineering and Biosciences, Instituto Superior Técnico, Universidade de Lisboa, Av. Rovisco Pais, 1049-001 Lisboa, Portugal; joaquim.cabral@tecnico.ulisboa.pt; 3Associate Laboratory i4HB—Institute for Health and Bioeconomy, Instituto Superior Técnico, Universidade de Lisboa, 1049-001 Lisboa, Portugal

**Keywords:** magnetic field, magnetic-responsive hydrogel, mesenchymal stromal cells, endothelial cells, VEGF, angiogenesis

## Abstract

This work shows the ability to remotely control the paracrine performance of mesenchymal stromal cells (MSCs) in producing an angiogenesis key molecule, vascular endothelial growth factor (VEGF-A), by modulation of an external magnetic field. This work compares for the first time the application of static and dynamic magnetic fields in angiogenesis in vitro model, exploring the effect of magnetic field intensity and dynamic regimes on the VEGF-A secretion potential of MSCs. Tissue scaffolds of gelatin doped with iron oxide nanoparticles (MNPs) were used as a platform for MSC proliferation. Dynamic magnetic field regimes were imposed by cyclic variation of the magnetic field intensity in different frequencies. The effect of the magnetic field intensity on cell behavior showed that higher intensity of 0.45 T was associated with increased cell death and a poor angiogenic effect. It was observed that static and dynamic magnetic stimulation with higher frequencies led to improved angiogenic performance on endothelial cells in comparison with a lower frequency regime. This work showed the possibility to control VEGF-A secretion by MSCs through modulation of the magnetic field, offering attractive perspectives of a non-invasive therapeutic option for several diseases by revascularizing damaged tissues or inhibiting metastasis formation during cancer progression.

## 1. Introduction

The formation of new blood vessels from existing ones (angiogenesis) is relatively rare in adults and occurs mostly in post-injury regeneration areas or during tumor growth [1]. Although these two types of angiogenesis are promoted by similar angiogenic signals, the application outcomes are very different. While regenerative angiogenesis repairs functional and interconnected vessels, tumor angiogenesis is characterized by a high number of immature and disorganized vessels [2,3].

Briefly, the complex mechanism of angiogenesis is initiated when increased concentration of pro-angiogenic factors—such as VEGF-A, angiopoietins, basic fibroblast growth factors (bFGFs), hepatocyte growth factors (hGFs), and platelet-derived growth factors (PDGF), among others—produced by inflammatory or tumor cells, activate quiescent endothelial cells from an existing vessel in response to injury and/or hypoxia. The activated cells differentiate into tip cells, creating elongated sprouts/new vessels toward the stimulus through active migration [3].

Pro-vascularization strategies have been explored to treat several diseases related to reduced vascular perfusion, such as tissue transplantation, peripheral vascular disease, ischemic heart disease, and wound healing [4]. Among the key features of the angiogenic mechanism are temporal regulation, the spatial organization of the stimuli, cellular crosstalk, active remodeling, and interaction with the extracellular matrix [5,6]. When these features are dysregulated, the development of new vasculature is abnormal (e.g., tumor angiogenesis). Anti-angiogenic procedures are also being exhaustively investigated mainly to reduce tumor growth in cancer patients [5,6].

Understanding the mechanisms regulating healthy and pathological angiogenesis is crucial to provide valuable information for the fabrication of tissue-engineered constructs embedded with robust and mature vasculature. 

Despite experimentally induced ischemia in animals having shown promising results, human trials using therapeutic vascularization have reported poor success rates [7]. Most proangiogenic strategies have mainly focused on the delivery of various isoforms of growth factors VEGF or even FGF in clinical trials [8,9]. One of the most promising approaches to provide continuous growth factor delivery to the injured tissues has been explored through scaffold functionalization [10,11,12]. Collagen coated with PDGF (platelet-derived growth factor) has shown enhanced capillary formation in a dermal wound healing model [10,11], while loading alginate or PLGA (polylactic-co-glycolic acid) scaffolds with VEGF was reported to promote vascularization [12,13,14,15].

However, the fast in vivo degradation of these molecules (~30 min) over time limits the direct intravenous administration of these growth factors [16,17], bringing to light the need to establish an external source of stimulation able to maintain a continuous release of growth factors to the in vivo environment injury site through a steady stimulation of the cells growing on the scaffold. 

To date, static magnetic fields have emerged with a strong potential in tissue engineering not only as a system to control the release of drugs, growth factors, and miRNA, but also by enhancing stimulatory responses able to influence cell behavior, mobility, and growth [18,19,20]. Moderate intensity magnetic fields (1 mT to 1 T) have been associated with new bone formation, decrease of mineral bone density, and enhanced metabolic activity in the repair of cartilage, while stronger magnetic fields (up to 8 T) have also induced bone formation by promoting matrix formation and osteoblast differentiation in vitro and in vivo [21,22]. Distinct magnetic-induced biological effects, under a dynamic regime, have been described in the literature. Dynamic magnetic fields revealed the capacity to promote morphological and intracellular changes driving cell differentiation into osteoblast-like cells [23,24]. However, dynamic magnetic field inhibitory effects on embryonic development and angiogenesis processes have also been reported [25]. A low-frequency magnetic field of 2 mT was found to inhibit angiogenesis processes by significantly reducing the VEGF signal transduction pathway on Human Umbilical Vein Endothelial Cells (HUVECs) [26]. Additionally, the direct exposure of Chick Embryo Chorioallantoic Membrane assay to a static magnetic field of 0.7 T and an electromagnetic field of 0.04 T resulted in angiogenic inhibition [27]. Other works described similar inhibitory effects of stronger intensity static magnetic fields (0.4–0.6 T) on vascularization and tumor growth in in vivo mice models and ex vivo experiments [28,29].

MSCs are known to improve angiogenesis behavior through the secretion of angiogenic growth factors such as VEGF-A [30]. The impact of magnetic field on the secretory activity of MSCs has been recently reported in a work from the same authors [31]. That work proved the sensitivity of MSCs to static magnetic fields, revealing that the magnetic field has the potential to exert a pro-angiogenic or anti-angiogenic effect depending on the chemistry of the scaffold. Furthermore, the magnetic field capacity to enhance cellular differentiation and increase intracellular free calcium concentration has also been described in the past [32,33].

As stated, the magnetic field can influence cellular processes; however, the regulation of these effects remains underexplored. Hence, this study aims to provide a better understanding of the capacity to non-invasively modulate the angiogenesis process in endothelial cells via the stimulation of VEGF-A production by MSCs, based on the variation of the magnetic field conditions, i.e., dynamic regimes and intensity. This study compares for the first time the impact of static and dynamic magnetic fields, applied at different intensities and variation regimes, on the secretion of angiogenic genes, mainly focusing on VEGF-A (key molecule in angiogenesis development) by MSCs cultured on magnetic-responsive gelatin scaffolds-mGelatin (Figure 1). Ultimately, this work envisages the use of MSCs as regulators/stimulators of vessel sprouting in vitro and highlights the therapeutic clinical translation potential of the combination of MSCs and magnetic field. The capacity of the magnetic field to regulate angiogenic events may be regarded as an immature strategy that may be prone to future development of magnetic-based therapies for the treatment of several vascular-related diseases. For instance, magnetic hydrogels loaded with MSCs may be injected in situ or used as an implantable stent to allow for the regeneration of damaged tissues and blood vessels through the overexpression of proangiogenic genes under non-invasive external magnetic stimulation. Finally, the possibility to downregulate angiogenesis with magnetic stimuli can also be clinically translated as a powerful vascularization inhibition tool for tumor metastasis during cancer progression.

## 2. Materials and Methods

### 2.1. Scaffold Fabrication

Magnetic nanoparticles (MNPs) were synthesized by chemical co-precipitation of iron salts FeCl_3_ and FeCl_2_ (Sigma-Aldrich, Saint louis, MI, USA) in alkaline media, according to the protocol published by Izquierdo et al. [34] Briefly, an aqueous solution with 25% of ammonium hydroxide (NH_4_OH) (Fluka, Munich, Germany) was added to a mixture of FeCl_3_ and FeCl_2_ at 80 °C, under permanent stirring at 1250 rpm by purging N_2_. Porcine skin gelatin (8% m/v, type A, G2500, Sigma-Aldrich) was dissolved in Milli-Q water at 60 °C. MNPs were dispersed by sonication in the polymeric aqueous solution. The mGelatin matrices were cast onto glass plates and left overnight at 4 °C. Finally, the samples were removed from the glass plates and immersed in an aqueous solution containing 1% of glutaraldehyde for 3 h. Before use, the magnetic hydrogel matrices were washed by immersion in a demineralized water bath. For sterilization, the samples were immersed in a PBS solution containing 1% of antibiotic-antimycotic (Gibco, New York, NY, USA). For better conditioning of the cells, the samples were immersed in cell culture media for 3 h before seeding the cells on top of the hydrogels.

### 2.2. Magnetic Profiles Characterization and Experimental Setup

Two neodymium magnets with different magnetic intensities (circular and square magnet bars allowing a maximum intensity of 0.08 T and 0.45 T, respectively—Figure 1) were used for cell stimulation. These magnets, containing high magnetocrystalline anisotropy, were characterized by their strength and magnetic homogeneity, with ability to magnetize along a specific crystal axis [35,36]. Despite the changes in shape, both magnets used in this work had the same composition and exhibited the same pattern regarding the magnetic orientation lines. Hence, we compared the bio-performance of these magnets when applied to MSCs cultures. To undertake a static continuous magnetic application on the cell culture, magnets were applied beneath the cell culture plate for 24 h, with the magnetic lines oriented perpendicularly to the scaffold surface/cell plate for the desired stimulation. For a dynamic setup, a motorized lab jack (MOZ-80-50, Optics Focus, Beijing, China) combined with a motion controller (For NEMA17, Optics Focus) was programmed to transport the magnets to the cell culture plate using different configuration modes. The motorized lab jack, with the magnet on top, was placed inside the cell incubator and positioned at a proper distance to allow for the magnet to reach the cell culture plate at the elevator maximal amplitude. As shown in Figure 1, a motion controller was connected to a computer and programmed for a motion velocity of 0.005 m/s for 24 h under the following conditions (Running Time = 1; Max Speed = 255; Delay = 0; Displacement = ±999999). For a lower frequency (LF) of 2.8 × 10^−4^ Hz, the magnet was kept 1 h at maximum displacement, followed by 1 h at a minimum displacement from the cell plate in continuous cycles for 24 h. For a higher frequency (HF) of 1.7 × 10^−2^ Hz, the magnet alternated between maximum and minimum displacement in cycles of 60 s, reaching the cell culture plate after 30 s, for a total exposure of 24 h (Figure 2). A gaussmeter was used to measure the intensity of the magnets in each point and build the magnetic profile of each condition mode. Each experiment was conducted for 24 h indicating that cells were exposed to a total of 1440 ON/OFF magnetic field cycles in HF experiments and 12 ON/OFF magnetic field cycles in LF experiments.

### 2.3. Cell Culture

MSC lines used in this work derive from human bone marrow samples, donated by healthy donors under informed consent, in accordance with Directive 2004/23/EC of the European Parliament and of the Council of 31 March 2004 on setting standards of quality and safety for the donation, procurement, testing, processing, preservation, storage, and distribution of human tissues and cells (Portuguese Law 22/2007, 29 June), with the approval of the Ethics Committee of the respective clinical institutions [37]. Three independent donors were used in the experiments (*n* = 3, donor A: male donor, 35 years old; samples isolated in 2015, B: male donor, 73 years old; samples isolated in 2008, C: male donor, 38 years old; samples isolated in 2015). These cells belong to the cell bank available at Stem Cell Engineering Research Group (SCERG), iBB—Institute for Bioengineering and Biosciences at the Instituto Superior Técnico (IST). Bone marrow samples were retrieved from the Instituto Português de Oncologia Francisco Gentil, Lisbon, Portugal under collaboration with iBB-IST. Isolated cells were cryopreserved in liquid/vapor nitrogen tanks. Isolated human bone marrow MSCs (BM MSCs) were cultured on low-glucose Dulbecco’s Modified Eagle Medium (DMEM, Gibco) supplemented with 10% fetal bovine serum (FBS MSC qualified, Gibco) and 1% antibiotic-antimycotic (Gibco) and kept at 37 °C, with 5% CO_2_ and 21% O_2_ in a humidified atmosphere. The phenotype of MSCs under magnetic exposure was confirmed in our previous study [31]. Briefly, the cells were tested by flow cytometry, with and without magnetic exposure, for expression of cell surface markers indicative of MSCs, using a panel of mouse anti-human monoclonal antibodies (PE-conjugated) against: CD73^+^, CD90^+^, CD105^+^, CD14^−^, and human leukocyte antigen HLA-DR^−^ (all from Biolegend, California, U.S.). All cellular experiments with MSCs were performed between passages 4 and 7. 

HUVECs used in this study are commercially available and were purchased from Lonza (Basel, Switzerland). HUVECs were expanded in commercial endothelial growth medium-2 (EGM-2, Lonza) and kept at 37 °C, with 5% CO_2_ in a humidified atmosphere. The medium renewal was performed every 3–4 days. All cellular experiments with HUVECs were performed between passages 4 and 7.

### 2.4. MSC Proliferation Values

MSCs proliferative assessment was evaluated 24 h after magnetic exposure (static or dynamic) and compared with the initial density of 75,000 cell/cm^3^. MSCs were seeded on top of mGelatin scaffolds at a density of using expansion medium (DMEM + 10% FBS). As a control, MSCs were seeded as a monolayer on cell culture plates at a density of 75,000 cell/cm^2^. The metabolic activity of MSCs was evaluated using AlamarBlue^®^ cell viability reagent (Molecular probes, Eugene, Oregon, U.S.), following manufacturer instructions. This non-toxic, cell-permeable reagent is a resazurin-based solution, blue in color and non-fluorescent, that functions as a cell health indicator by using the reducing power of living cells to quantitatively measure viability. MSCs treated with 10% (*v/v*) AlamarBlue^®^ cell viability reagent were incubated at 37 °C in a 5% CO_2_ chamber for 2 h. After entering the cells, the resazurin compound of AlamarBlue^®^ is reduced to resorufin, a compound red in color and highly fluorescent. MSCs treated with 10% (*v/v*) AlamarBlue^®^ cell viability reagent were incubated at 37 °C in 5% CO_2_ chamber for 2 h. The fluorescence intensity of the supernatant of the cells was quantified in a range of 560–590 nm. Prior to analysis, a calibration curve for different human bone marrow MSC densities (10,000; 20,000; 50,000; 75,000; 100,000; 150,000 cells/mL) was used to convert the obtained metabolic values into cell numbers and determine the cell proliferation values associated with each group of conditions. All conditions were tested in triplicates during this experiment using three independent MSC donors (*n* = 3).

### 2.5. CD31 Immunostaining

HUVECs immune characterization was carried out using CD31 antibody after cell culture fixation with 4% paraformaldehyde and blocking with 10% FBS (Fetal Bovine Serum, Gibco) in PBS solution. Primary antibody CD31 (1:50 dilution, mouse antibody, Dako, Santa Clara, CA, USA) in block solution was incubated overnight. Secondary antibody Alexa Fluor 546 (1:500 dilution, goat anti-mouse, Abcam, Cambridge, United Kingdom) was added and incubated for 30 min. Images were obtained using a fluorescence microscope (LEICA DM IL LED with EC3 camera system) and a confocal laser scanning microscope (LSM 700/ Carl Zeiss, Jena, Germany).

### 2.6. Cell Viability and Morphology Assay

MSCs were seeded on the scaffolds at a density of 75,000 cell/ cm^3^. To assess the MSCs’ viability, cells were washed twice with PBS (Phosphate Buffer Saline, Dulbecco’s Sigma-Aldrich) after 24 h of magnetic application and stained for 20 min using LIVE/DEAD^TM^ Viability/Cytotoxicity Kit, containing calcein and ethidium homodimer probes (L3224, ThermoFisher Scientific, Waltham, MA, USA), according to the manufacturer protocol for mammalian cells. After washing with PBS, cell imaging was obtained using a fluorescence microscope (Leica DM IL LED with EC3 camera system, Wetzlar, Germany). The external membrane of living cells was stained with calcein probe and observed in a green coloring while dead cells were highlighted with a red nucleus by ethidium homodimer probe. The calcein staining of the membrane of the cells also allows for a morphological analysis of the cells under different experimental conditions.

### 2.7. Quantitative Reverse Transcription Polymerase Chain Reaction Analysis

MSC RNA was extracted using a RNeasy Mini Kit (Qiagen, Hilden, Germany). Complementary DNA was synthesized from 20 ng of total RNA using iScript Reverse Transcription Supermix (Bio-Rad, Hercules, CA, USA). The reaction mixture (20 µL) was incubated in a thermal cycler (Veriti 96-well thermal cycler: Applied Biosystems, Foster City, California, U.S.) for 5 min at 25 °C, 30 min at 42 °C, and 5 min at 85 °C. Samples were then kept at 4 °C. Gene expression levels of VEGF, bFGF, and hGF were assessed. Quantitative reverse transcription-polymerase chain reaction (qRT-PCR) was performed using SYBR Green PCR Master Mix (Applied Biosystems). All reactions were carried out at 95 °C for 10 min, followed by 40 cycles of 95 °C for 15 s and 60 °C for 1 min, according to the manufacturer’s instructions. All groups of conditions were performed in triplicates in the experiment (*n* = 3) using three different donors (donors A, B, C). Glyceraldehyde 3-phosphate dehydrogenase (GAPDH) was used as host gene control to normalize differences in total RNA levels for each condition. A threshold cycle (C*t*) was observed in the exponential phase of amplification, and the quantification of relative expression levels was performed using standard curves for the target genes and the endogenous control. Geometric means were used to calculate the △△C*t* values and expressed as 2^△△C*t*^ (Potency values). The mean values from the triplicate analysis were compared.

### 2.8. Quantification of VEGF Expression by Enzyme-Linked Immunosorbent Assay (ELISA)

Culture supernatant (secretome) was collected from the MSCs cultures after 24 h of magnetic exposure (mGelatin and monolayer). The secretome used as conditioned media was kept at −80 °C until further analysis. A human VEGF-A kit (RayBiotech, Georgia, U.S.) was used, following the manufacturer’s instructions. Briefly, 100 µL of conditioned media from MSCs culture exposed to different magnetic conditions was added to the ELISA 96 well-plate coated with the antibody specific for human VEGF-A. The samples were incubated for a total amount of 255 min, including gentle shaking and several washing procedures, and finally, the intensity of the color was measured at 450 nm. All conditions were tested in triplicates using three different donors (donors A, B, C).

### 2.9. In Vitro Endothelial Cell Tube Formation Assay

To evaluate the effect of exogenous VEGF-A supplementation on angiogenic properties, a three-dimensional capillary-like tube formation assay was performed. For this experiment, MSCs were incubated with Endothelial Cell Basal Medium (EBM-2, Lonza), VEGF-free media, during magnetic application. Simultaneously, HUVECs (2 × 10^4^ cells) were cultured on Matrigel (50 µL/well) in a 96-well plate. The conditioned media (without VEGF and FBS supplements) obtained from MSCs, cultured on scaffolds or in monolayer under magnetic effect, was added to each well. The observation of the induced HUVECs sprouting on the conditions of the analysis was compared with positive and negative controls for this experiment, i.e., the controls, where HUVECs were supplemented with Endothelial Cell Growth Media-2 (EGM-2), rich in growth factor supplements, to promote HUVECs sprouting (positive control) and controls where the secretome from MSCs was grown in polystyrene without magnetic exposure to assess the effect of the residual secretion of growth factors by MSCs that were never exposed to magnetic stimulation (negative control). After incubation for 6 h at 37 °C, three images were taken from the center of each well using a light microscope (Leica DM IL LED with EC3 camera system). The number of tubular-like structures and branch points formed was counted using ImageJ (NIH) software. All conditions were tested in triplicates using three different donors (donors A, B, C).

### 2.10. Statistical Analysis

All measurements were performed three times, under independent conditions. Results were presented as the mean ± standard deviation (SD). Two-way ANOVA Sidak’s multiple comparisons test was used to compare the mean of three values obtained from three independent conditions, using GraphPad Prism version 7 (GraphPad Software, La Jolla, California, U.S.); **p* < 0.05 indicates a significant result; ***p* < 0.01 a very significant result, ****p* < 0.001 a highly significant result; and *****p* < 0.0001 an extremely significant result.

## 3. Results

### 3.1. Effect of the Magnetic Field on the Toxicity and Morphology of Mscs

The potential toxicity of the magnetic field on MSCs cultured on mGelatin scaffolds and cell polystyrene plates (TCP) was evaluated and assessed by analysis of the proliferation values of MSCs when exposed to a constant magnetic field (static mode) at 0.08 T and 0.45 T and to a cyclic variation of the magnetic field (dynamic mode), changing between 0 T and 0.08 T or 0 T and 0.45 T.

Two different dynamic conditions were considered: (1) obtained by a low-frequency variation of the magnetic field (LF), involving the exposure of MSCs to subsequent variation of magnetic bar position each 30 min (Figure 2a) and (2) obtained by high-frequency variation of the magnetic field (HF), where MSCs were subjected to uninterrupted variation of the magnetic field bar position (Figure 2b).

Cell proliferation studies showed that the increase of the magnetic field intensity from 0.08 T to 0.45 T led to a decrease of the number of cells, observed either for MSCs cultured in mGelatin scaffolds as well as for MSCs monolayers cultured in polystyrene cell plates (negative control). A comparative analysis of the effect of the magnetic field applied under the different dynamic regimes (LF and HF dynamic regimes) revealed that LF dynamic mode resulted in a lower number of cells than HF or static modes. This result suggested that the LF dynamic mode might induce higher disruption of the cells and irreversible oxidative stress leading to increased cell death, as a common effect of the magnetic forces on cells [38,39,40].

Live/Dead assays were performed to investigate cell death and the cell morphology induced by each magnetic field condition tested. Live/Death results (Figure 3a) did not show cell death for MSCs cultured on mGelatin scaffolds in any magnetic field condition. It was observed that magnetic field induced the alignments of the MSCs. However, the alignment did not show any dependence on the magnetic field intensity (0.08 T and 0.45 T) or dynamic regime. No significative cell patterning alterations were registered for MSCs exposed to static, LF, or HF dynamic regimes, except for the lower MSC density observed upon exposure to LF conditions (Appendix A).

Nonetheless, it was curious to observe that the response of MSCs cultured in TCP to magnetic field was substantially different. In this case, cell death was only detected in MSCs cultures exposed to a static 0.45 T magnetic field (Figure 3a), whereas an irregular cell distribution, denoted by the presence of empty regions (regions without cells), was observed for MSC cultures exposed to dynamic regimes at higher magnetic field intensity (0.45 T). Furthermore, MSCs showed the capacity to align when exposed to static and HF dynamic magnetic field, but poor cell orientation when exposed to LF dynamic regimes (Figure 3a). 

### 3.2. Impact of the Magnetic Field on the Expression of VEGF-A Gene

The dependence of MSCs behavior on the magnetic intensity and dynamic conditions, described in the previous section, suggested the possibility to regulate cellular pathways in MSCs, in particular those concerning specific gene expression, by modulation of an external magnetic field. Hence, studies were performed to evaluate the effect of the different magnetic field intensities and dynamic regimes on the MSC capacity to express the angiogenic growth factor gene, VEGF-A. Other angiogenic genes were also analyzed (bFGF and hGF) alongside VEGF-A for the static magnetic regime under 0.08 T and compared with the respective condition not exposed to magnetic stimulation—0 T (Appendix A). However, the magnetic field did not show an impact on the expression of bFGF and hGF angiogenic genes by MSCs, revealing no differences between the conditions 0.08 T and 0 T. For this reason, our study was mainly focused on the magnetic field effect on VEGF-A expression and secretion.

The influence of the different magnetic field conditions on VEGF-A expression by MSCs cultured on mGelatin scaffolds and in TCP (monolayer MSCs) is represented in Figure 4.

The results revealed that VEGF-A expression was affected by both the magnetic field intensity and the cyclic dynamics. VEGF-A expression showed a clear decrease with the increase of magnetic field intensity to 0.45 T, at all the magnetic field dynamic regimes tested. This effect was observed either for MSCs cultured in mGelatin scaffolds as well as for MSCs monolayers cultured in TCP (negative control). 

The VEGF-A expression by MSCs cultured in mGelatin scaffolds was found to be higher at static magnetic field than at dynamic magnetic field conditions, suggesting that VEGF-A expression was inhibited by the latter regimes. As shown in Figure 4, the decline of VEGF-A expression was more substantial for MSCs exposed to LF than to HF magnetic regimes, suggesting that the negative effect of magnetic cyclic conditions correlates with the extension of the intervals in the absence of magnetic field (magnetic field OFF). These results are in good agreement with the negative effect of the magnetic field on cell proliferation, as the reduced VEGF-A expression may be ascribed with a lower number of cells under dynamic field conditions. Additionally, these results are compatible with the Live/Dead assays, indicating that the inhibition of VEGF-A expression may be related to the lower cell homogeneity and the lower magnetic responsiveness (i.e., poorer magnetic cell alignment) found for MSCs cultures under LF cyclic regimes. A similar trend was observed for VEGF-A expression by the MSC monolayer cultured in TCP. However, in this case, the differences in VEGF-A expression from MSCs exposed to static and HF magnetic field regimes were negligible.

### 3.3. Control of VEGF-A Protein Secretion under Magnetic Exposure

The expression of the VEGF-A gene on MSCs was quantified in the previous section. However, this value reflects the capacity of MSCs to express VEGF-A, which does not necessarily correspond to the amount of VEGF-A present in the external conditioned cell media (VEGF-A secretion), fundamental to induce angiogenic behavior in HUVECs.

This section evaluates the ability to magnetically regulate VEGF-A expression by MSCs based on the amount of VEGF-A released by these cells to the extracellular media. VEGF-A was quantified by ELISA (Figure 5). 

The secretion of VEGF-A by MSCs showed to be significantly influenced by the magnetic field conditions applied, supporting the conclusions obtained by analysis of VEGF-A gene expression (in the previous section) and highlighting once more the importance of establishing the adequate magnetic field strategies which may allow for improved regulation of cell behavior. Similar to the analysis of VEGF-A gene expression, a clear reduction of VEGF-A secretion by MSCs under stronger magnetic intensity (0.45 T), coherent for all tested donors, was observed.

The effect of magnetic field dynamics on VEGF-A expression was confirmed by VEGF-A quantification in the MSC extracellular media. Comparable VEGF-A secretion was obtained under static and HF dynamic regimes at 0.08 T, whereas LF magnetic field cycles inhibited the production of VEGF-A by the MSCs. 

### 3.4. Magnetic Effect on the Sprouting Potential of HUVECs

The conditioned extracellular media produced by MSCs (MSCs secretome) was expected to induce angiogenesis on HUVECs through the formation of tube-like structures proportionally to the concentration of growth factors, such as VEGF-A, in the media (Figure 6). The magnetic stimulation of VEGF-A expression by MSCs should then be followed by an improved HUVEC angiogenic potential.

The impact of the magnetic field on the angiogenic potential was ultimately studied based on the ability to magnetically modulate the sprouting effect on endothelial cells (HUVECs) using the MSC secretome. 

As shown in Figure 6a,b, functional assays performed using the conditioned cell media obtained at higher magnetic field intensities (0.45 T), from MSCs cultured in mGelatin scaffolds, led to a clear reduction in the number of tubes and branch points (below 40 tubes and 20 branch points, reaching the minimum amount of 5 tubes and 0 branch points under LF regime). In comparison with the negative control (43 tubes and 28 branch points), performed using the secretome from MSCs cultured in TCP without magnetic stimulation, the number of tubes formed under the magnetic intensity of 0.45 T were significantly lower (Figure 6b,c), thus evidencing the anti-angiogenic potential of high magnetic field intensity (0.45 T). 

Contrastingly, the use of the secretome obtained from MSCs exposed to static magnetic field conditions at 0.08 T led to a more intense sprouting effect, resulting in the formation of the highest number of tubes (80 tubes on average) and branch points (48 branch points). 

It is important to highlight that HUVEC sprouting obtained with the secretome produced by MSCs exposed to the static magnetic field was higher than that obtained with the negative control and slightly higher, in terms of tube formation, than that observed for the positive control condition obtained using endothelial cell-specific media, rich in growth factor supplements, such as VEGF-A (75 tubes and 64 branch points). These results revealed the angiogenic potential of this magnetic field strategy, i.e., static magnetic field of low intensity. 

The anti-angiogenic effect of high-intensity magnetic field was less clear when using the secretome from MSCs cultured in TCP. The number of tubes formed with the secretome from MSC monolayers exposed to dynamic magnetic field regimes at 0.45 T was lower than that observed with the secretome of MSC monolayers exposed to a dynamic magnetic field at 0.08 T. However, the opposite effect was observed for experiments conducted using the secretome related with the exposure of MSC monolayers to a static magnetic field.

The results also revealed differences in HUVEC sprouting associated with the magnetic dynamic regimes used. The formation of HUVEC microvessels was reduced in experiments using the secretome from MSCs exposed to dynamic magnetic field regimes in comparison with the results associated with static magnetic field conditions. The lowest HUVEC sprouting effect (12 tubes and 8 branch points in the monolayer condition and 5 tubes and 0 branch points in the scaffold condition) was observed with the secretome from MSCs exposed to LF magnetic field dynamic regimes. Similar results were obtained for functional assays using the conditioned media from MSCs monolayers cultured in TCP.

Overall, these results showed the possibility to effectively regulate angiogenesis, remotely, by selecting the magnetic field conditions responsible for triggering MSCs into secreting desirable levels of VEGF-A to prompt HUVEC maturation. A monolayer of HUVECs culture was marked with CD31 antibody (specific endothelial cell marker), confirming the endothelial lineage of HUVECs and ability for sprouting and tube formation (Appendix A). Cell viability of HUVECs, prior to tube formation assay in monolayer culture and during sprouting, was also assessed using Calcein probe. The images allowed to identify the living cells and evaluate a high cell survival during the process of vessel branching, as well as the morphological differences between the cells in monolayer and the formation of vessel branching structures (Appendix A).

Angiogenesis can be upregulated by exposing MSCs cultures to static or HF magnetic fields with low intensity, whereas angiogenesis downregulation may be attained by simply increasing the magnetic field strength to values of 0.45 T (keeping the dynamic regime) or by imposing a LF magnetic field cycle regime while maintaining a low magnetic field intensity. Combined, these results might contribute to the development of a controlled non-invasive magnetic responsive platform capable of enhancing vascularization in damaged tissues or to inhibit vascularization during tumor progression.

## 4. Discussion

The magnetic strength combined with the homogeneity of neodymium magnets has potentiated its usability for medical applications, in particular in magnetic resonance imaging, as an alternative to superconducting magnets, which required a coil of superconducting wire to create a magnetic field [41]. Nowadays, these magnets are also surgically implanted around the lower esophageal sphincter to treat gastroesophageal reflux disease and for the treatment of sensory insensibility [41,42]. The medical interest in the use of neodymium magnets has triggered the investigation of the impact of different magnetic field parameters, such as intensity and application modes at the cellular level. Therefore, the present study investigates the indirect impact of a magnetic field created by neodymium magnets on the ability of MSCs to produce and secrete VEGF-A, which in turn potentiates a modulatory effect on the angiogenic behavior of HUVECs, ultimately leading to the formation of microvessel structures. 

The potential to regulate the angiogenic behavior on HUVECs can be associated with homeostatic feedback mechanisms and represents a key for controlling healthy (e.g., tissue regeneration) or pathological angiogenesis (e.g., tumor development) while contributing to the treatment of several diseases. Hence, understanding the possibility to establish control over the secretion of VEGF-A from MSCs by modulation of external magnetic field stimulatory conditions might be the required answer to regulate the angiogenesis phenomena. 

Low-intensity magnetic field (0.02–0.12 T) has been described to affect the growth and development of MSCs [33]. Different works have been published reporting the influence of the magnetic field as a regulator of the ionic concentration within the cytoplasm, as a promoter of MSCs differentiation, and the ability to impact on cell morphology by modifying the polarization of cellular components [33,43,44]. The effect of a direct application of the magnetic field on endothelial cells has been mostly reported as an active inhibitor of sprouting and angiogenic activity [27,28,29,45]. However, the present work proves that the magnetic field may also exert a positive effect, allowing for indirect stimulation of angiogenic events on endothelial cells, e.g., HUVECs. As shown, the angiogenic activity of HUVECs may be regulated through magnetic modulation of the MSCs’ capacity to express and secrete VEGF-A. Pro-angiogenic or anti-angiogenic activities may be controlled by supplementing HUVECs with the conditioned media (secretome) produced by MSCs under variable magnetic field conditions. VEGF-A expression was stimulated at magnetic field intensities as low as 0.08 T and by exposure of MSCs to permanent magnetic field conditions (static magnetic field) but inhibited when stronger magnetic field intensities (0.45 T) are used or by exposure of MSCs to low-frequency magnetic field cycles (LF magnetic field). The results for static magnetic field are also supported by the conclusions from a previous study [31]. In that study, the secretion of VEGF-A by MSCs in the presence and absence of the magnetic field was compared, highlighting the impact of the magnetic field in the increased production of the angiogenic molecule [31].

In the present work, the magnetic modulation of the angiogenic molecule expression was reflected in the capacity of HUVECs to form tube-like structures and vessel ramifications developed in the presence of the respective MSC conditioned media obtained under different magnetic field conditions. Higher tube formation and vessel sprouting were observed for HUVECs developed in conditioned media obtained from MSCs exposed to static magnetic field of low intensity (pro-angiogenic effect). In contrast, the maturation of HUVECs with the secretome of MSCs exposed to LF dynamic modes or stronger magnetic field intensities resulted in a weaker angiogenic effect on HUVECs with reduced tube formation and branching (anti-angiogenic effect). It is important to highlight that the pro-angiogenic effect of the static magnetic field was able to slightly surpass the tube formation observed for HUVECs in positive controls. Similarly, the anti-angiogenic effect induced by LF magnetic field resulted in lower tube formation than that obtained for MSCs cultured in TCP cell plates without magnetic field stimuli (negative control). These results prove that the magnetic field may effectively regulate angiogenesis, promoting a pro-angiogenesis and anti-angiogenesis effect solely depending on a proper selection of the magnetic field parameters, i.e., magnetic field intensity and/or magnetic field variation dynamics. This regulatory capacity of the magnetic field on the cells may be attributed to its impact on MSCs proliferation and viability.

Effectively, the modulatory effect of the magnetic field on cell behavior was also highlighted in the present work by cell proliferation and viability assessment, which provided insights into the distinctive effect of high and low magnetic field intensities and magnetic field dynamic conditions on cell toxicity and organization (Figure 3). Cell proliferation studies showed that high-intensity magnetic field led to a decrease in the number of adherent cells, which suggests a potential effect of magnetic field on cell detachment or cell death [46], proportional to the magnetic field intensity (Figure 3a). The hypothesis of cell detachment to explain the negative effect of the magnetic field in the number of adherent cells is in good agreement with the results reported in previous work [47] from the same authors, which shows the impact of the magnetic field in the decrease of protein sorption in magnetic hydrogels. This effect was partially ascribed to a reduced protein attachment (or protein detachment) caused by the increased surface wettability in the presence of the magnetic field. Since cell attachment to scaffolds is mediated by protein interactions, an identical effect may also explain the cell detachment observed in the present work. Although the effect of the magnetic field was observed for MSCs cultured in mGelatin scaffolds and TCPs (in the absence of scaffold), the decrease in cell number is more notorious for MSCs in mGelatin scaffolds than in TCPs, when exposed to magnetic field intensity of 0.45 T, supporting this hypothesis.

Concomitantly, different cell number values were registered when changing the magnetic cyclic regimes. Lower cell proliferation was observed by decreasing the frequency of the magnetic field cycles (LF magnetic field), suggesting a possible inhibitory effect associated with the periodic intervals without magnetic interference (Figure 3b).

Cell toxicity showed also to be dependent on the magnetic field conditions, as it was higher at increased magnetic field intensity (0.45 T). Hence, the higher cell apoptosis at 0.45 T together with the eventual influence of magnetic field in VEGF-A expression regulatory cellular pathways may explain the reduced levels of VEGF-A expressed by MSCs exposed to such high magnetic field intensities (Figure 4) and the poor tube formation observed for HUVECs fed with the secretome from MSCs exposed to these magnetic field conditions (anti-angiogenic effect).

On the other hand, it is known that VEGF-A secretion contributes to an anti-apoptotic effect on the cells [48,49]. Therefore, a balance is expectable between the apoptotic effect caused by the magnetic field and the anti-apoptotic influence triggered by VEGF-A expression to maintain a healthy cell culture.

The magnetic response of MSCs in mGelatin scaffolds and TCPs showed similar trends. Tube formation and vessel sprouting were highly stimulated in HUVECs cultured with the secretome of MSCs on mGelatin under static magnetic field conditions. However, when the magnetic field was applied continuously in a static regime, the secretion of VEGF-A by MSCs cultured in mGelatin exceeded the values obtained with MSCs in TCP (Figure 4). It has been reported in the literature that cells directly exposed to external stressors, such as the magnetic field (e.g., the cells cultured in TCP), are prone to undergo cellular adaptation to stress, resulting in an increased release of survival growth factors (e.g., VEGF-A) during that period [50,51]. This seems to imply that MSCs in TCP might react faster to the cyclic magnetic stimuli, and undergo faster adaptation in comparison with MSCs on mGelatin scaffolds, where the magnetic stimuli were also influenced by the magnetic content of the hydrogel. In this regard, it is important to highlight that the superparamagnetic nanoparticles [52,53] in the mGelatin are described to have the ability to create single magnetic microdomains [54] within the flexible hydrogel. These magnetic microdomains might be responsible for different magnetic responses on the cells cultured on the mGelatin, due to either the hydrogel network or topographic impact [55,56], depending on the magnetic field configurations (static or dynamic). This might justify the differences observed for MSCs cultured on magnetic-responsive scaffolds and MSCs cultured on TCP. Additionally, these differences might also be partially ascribed to the bio-affinity of MSCs to the scaffolds and the culture plates, in particular due to the protein-binding receptors for MSCs on the mGelatin [46].

Regarding the interpretation of the results, the magnetic field effect on VEGF-A expression and angiogenesis cannot be explained by the average magnetic intensity or the total exposure time to the cells. While the average magnetic field intensity on the MSCs using the HF mode was lower (0.0013 T and 0.0073 T) than static 0.08 T intensity, the LF mode using the 0.08 T magnet provided an average magnetic intensity of 0.045 T. Yet, the angiogenic results were significantly weaker under the LF mode compared to the static mode. Instead, the different modulatory effects found for high- and low-frequency magnetic field dynamics may be related to the possible influence of the relaxation time of the magnetic field on cell behavior, again reporting the concept of cell adaptation to external stimuli. Cell adaptation has been rarely discussed in the literature. Nonetheless, a particular study reports how electrophysiological responses of olfactory cell receptors exhibited cell adaptation to step and multi-pulsed stimulation in a homeostatic feedback regulation [57]. This exposed the fact that the cells might require an adaptation period after pulsed frequencies. That may not be the case for the short-pulsed mode (HF) since the magnetic exposure might be too short (1 s) to require cell adaptation, bringing the HF mode very close to the static mode. The same is not valid for the LF mode, where the cells remain exposed to the magnetic field for 1 h and in the absence of the magnetic field for an identical period. In this case, the relaxation period may be followed by cell magnetic relaxation effect or higher cell stress release resulting in a lower pro-angiogenic effect. Another consideration that should be emphasized regarding the dynamic regimes is the Maxwell–Faraday Law, which states that a time-varying magnetic field is always responsible for a spatially varying non-conservative electric field resulting in electromagnetic induction with the generation of current [58,59]. While a static magnetic field does not induce currents in stationary objects, since there is no variation over time, and thus, does not have an associated frequency, in the case of HF regimes, where the magnetic field is constantly changing in time, the Maxwell–Faraday Law consideration is particularly relevant. It indicates that the cells exposed to this condition are not only experiencing the effects of the magnetic field but also of an electrical field, described as highly interfering with biomolecule and ion interactions in cell surface recognition mechanisms [60,61].

This report highlights the capacity to remotely regulate angiogenic behavior through the manipulation of magnetic dynamic regimes. The results evidenced the potential for future clinical translation with the development of novel magnetic-based therapeutic routes for several diseases, for instance, to use the LF magnetic regime for anti-tumorigenic applications for cancer treatment, aiming to decrease tumor vascularization and delay cancer progression. Likewise, the condition using a mGelatin scaffold and static low-intensity magnetic field indicated enhancement of the angiogenic potential with possible applications to revascularize damaged tissues in several vascular diseases (e.g., vascular diseases, diabetes, and atherosclerosis). Additionally, the potential impact of the magnetic stimulation on other molecules from the VEGF family, such as VEGF-C, might also offer perspectives on the application of this approach to promote lymphangiogenesis for the treatment of lymphatic conditions.

Despite these promising results, more exhaustive investigation would be required for a deeper and more complete understanding of the potential effect of magnetic field on other cell mechanisms involved in angiogenesis. Testing new donors, to decrease the effect of donor variability, and exploring the impact of the magnetic field over the secretion of other molecules by MSCs might also contribute to the robustness of the study and the validation of the clinical applicability of this system.

## 5. Conclusions

This work investigates the indirect impact of magnetic field on the angiogenic activity of HUVECs. The study was based on the ability to regulate VEGF-A expression by MSCs through changes in the intensity and dynamic regime of the applied magnetic field.

Lower magnetic field intensity (0.08 T), as well as a high-frequency magnetic variation regime, resulted in the increased production of growth factor VEGF-A compared to that observed at a magnetic field with a higher intensity (0.45 T) or applied under low frequency modes (LF modes), which translated into a positive angiogenic effect in HUVECs with the formation of a high number of tubes and branch points. A higher magnetic intensity of 0.45 T showed a significant decrease in the angiogenic potential of MSCs, consistent with the observed decreased cell number, suggesting increased mortality of MSCs when exposed to higher magnetic intensities. Lower frequency cyclic magnetic field regimes were also associated with an anti-angiogenic performance, demonstrated by the decreased secretion of VEGF-A in MSCs and by the lower number of tubes formed by HUVECs. Despite the better angiogenic response of MSCs in the presence of mGelatin, MSC monolayers (control) also demonstrated a satisfactory magnetic response, particularly evident under dynamic magnetic field regimes. From the clinical translational point of view, this work proves that the magnetic regulation of angiogenesis may be attained non-invasively in the presence and in some extent in the absence of magnetic support (magnetic-responsive gelatin scaffold), i.e., accounting with the magnetic susceptibility of MSCs. Overall, these results reveal that magnetic stimulation may either function as a tool to reduce tumor growth in association with cancer treatments (using LF operation modes) or as a possibility to promote the regeneration of diseased blood vessels (static and HF modes) in a contribution to vascular diseases treatments through cellular therapies. Although complementary studies are still required for a deeper understanding of the cell mechanisms, the present investigation reveals a modulatory effect of magnetic field regimes on MSCs behavior, thus offering the possibility to regulate angiogenesis phenomena with a potential contribution to future clinical applicability.

## Figures and Tables

**Figure 1 polymers-13-01883-f001:**
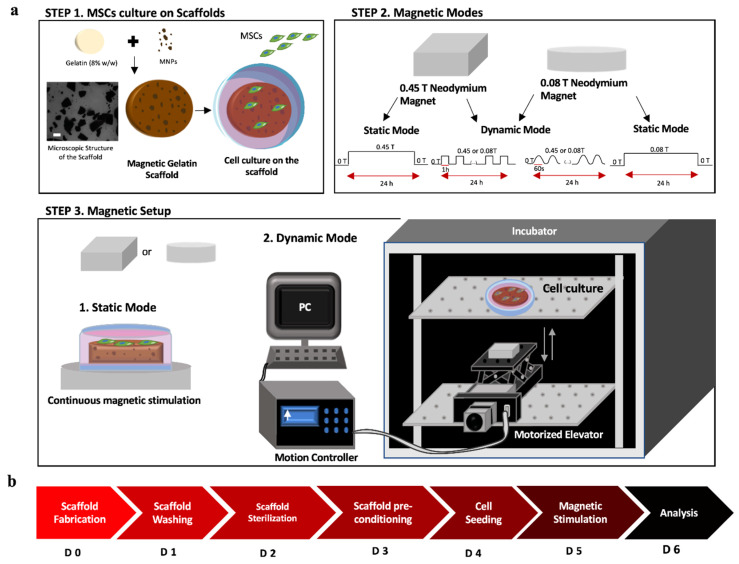
(**a**) Schematic setup for static and dynamic magnetic field modes using neodymium magnets of different intensities. Step 1: Scheme of the scaffold components (gelatin 8% *w/w* and MNPs), microscopic image of the MNPs distribution on the scaffold (scale bar: 100 µm), and MSCs cell culture on top of the mGelatin. Step 2: Configurations of the magnetic variation cycles and frequencies used in each condition (dynamic and static mode). Step 3: Setup of the position of the magnets on the bottom of the culture plate (1) and (2) configuration of magnets on top of the lab jack elevator to stimulate the cell culture inside the incubator. (**b**) Timeline for the stages of the experiment, from the scaffold fabrication to the readouts after the magnetic exposure of the MSCs.

**Figure 2 polymers-13-01883-f002:**
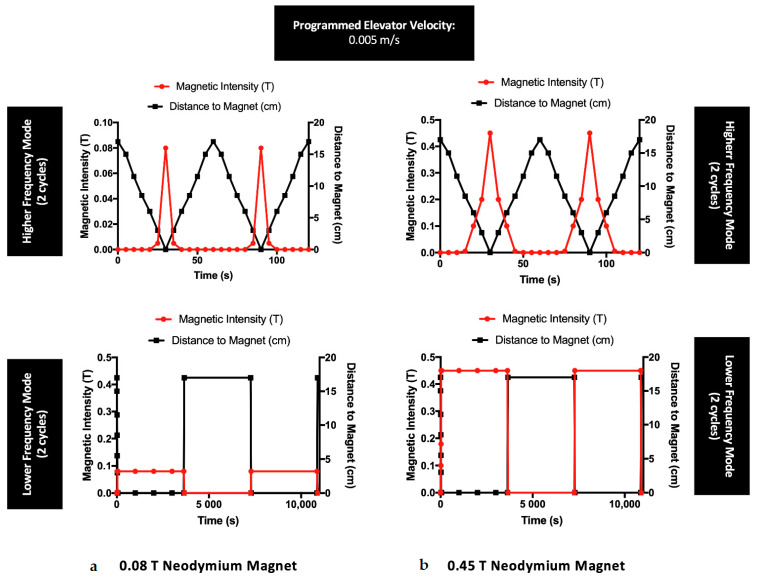
Magnetic profiles of the neodymium magnets (with a maximum intensity of (**a**) 0.08 T and (**b**) 0.45 T) used for the experiments with MSCs exposed to dynamic regimes, under a steady velocity of 0.005 m/s. Each frequency is represented with two cycles concerning the distance between the magnets and the cell culture plate, during a total magnetic exposure of 24 h.

**Figure 3 polymers-13-01883-f003:**
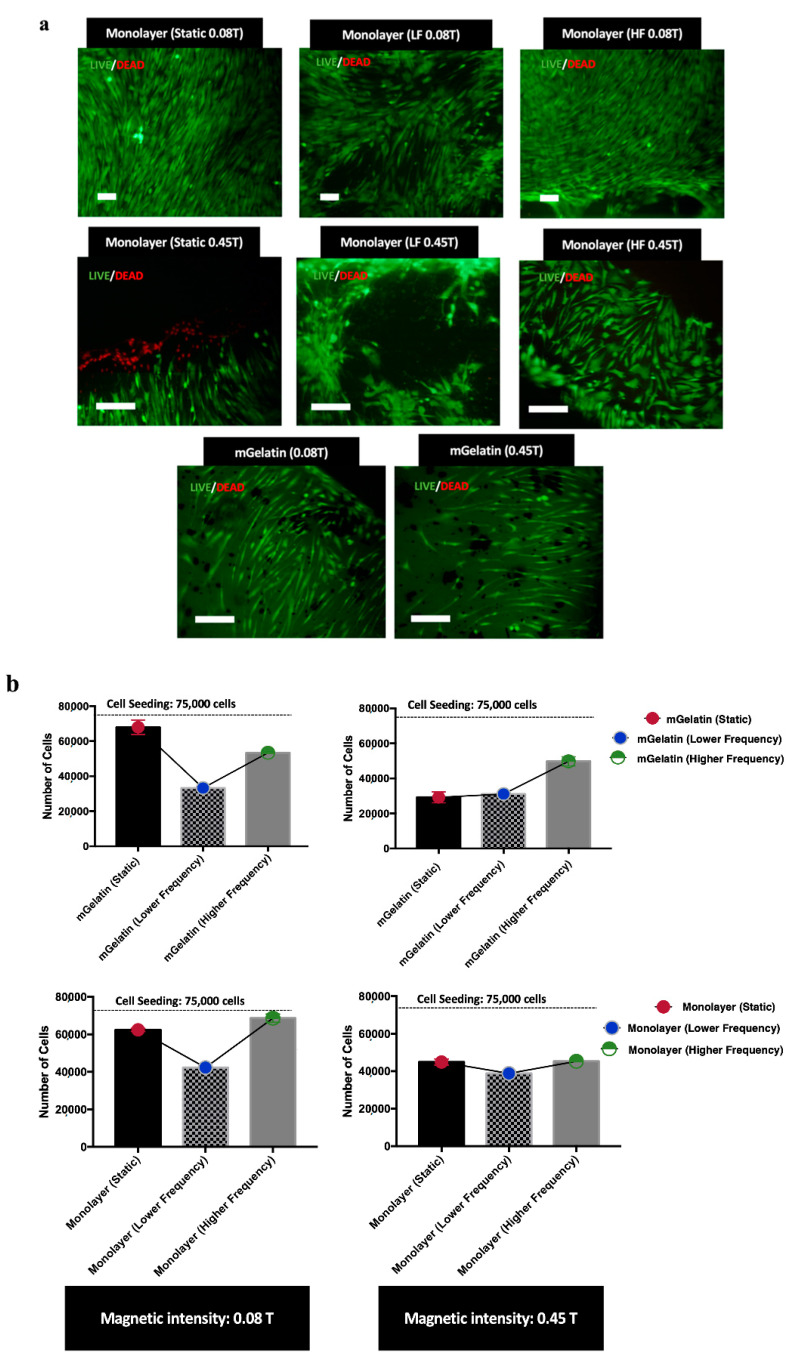
(**a**) Live (Green) and Dead (Red) cells images MSCs in monolayer for each condition, highlighting distinct cell orientation profiles and morphologies depending on the magnetic field conditions. MSCs distribution on mGelatin is also represented for the lower and higher maximum intensity (0.08 T and 0.45 T). Scale bar: 100 µm. (**b**) Proliferation values for MSCs cultured for 24 h on mGelatin scaffolds and MSCs monolayer under exposure to static and dynamic magnetic fields with 0.08 T and 0.45 T. Data are presented as means ± SD.

**Figure 4 polymers-13-01883-f004:**
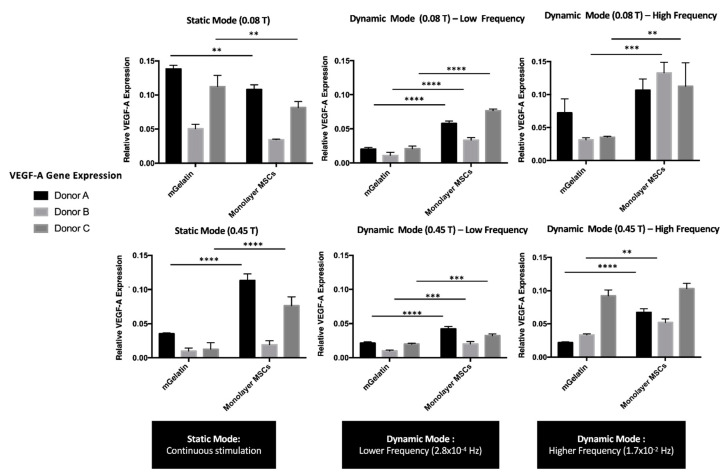
VEGF-A expression values using RT-PCR technique to compare the effect of static and dynamic modes (low and high frequency magnetic field) and magnetic intensities (0.08 T and 0.45 T) over MSC secretome. Relative expression of VEGF-A from MSCs cultured on mGelatin scaffolds, in comparison with monolayer MSCs as a control condition, using three MSC donors (A, B, C) under independent conditions. Data are presented as means ± SD. Statistical significance was determined using two-way ANOVA: ***p* < 0.01 a very significant result, ****p* < 0.001 a highly significant result; and *****p* < 0.0001 an extremely significant result.

**Figure 5 polymers-13-01883-f005:**
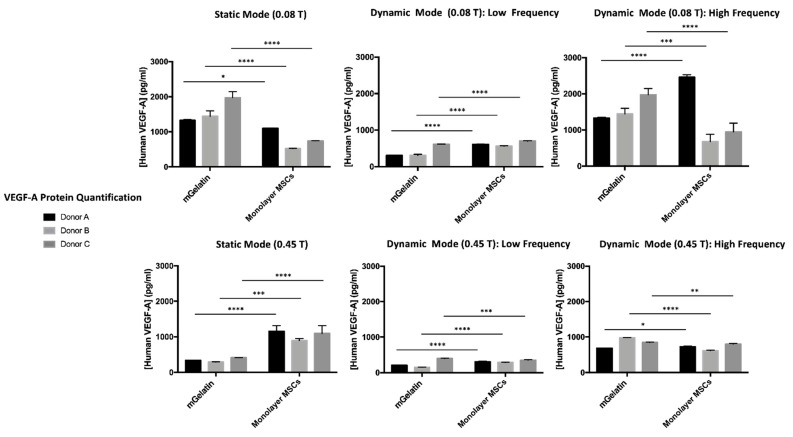
VEGF-A protein quantification for MSCs cultured on mGelatin and control condition (MSCs cultured on monolayer) for each experimental condition (static, lower, and higher frequency magnetic field) and corresponding donor (A, B, C), according to the magnetic intensity (0.08 T and 0.45 T). Three MSC donors (A, B, C) were used under independent conditions. Data are presented as means ± SD. Statistical significance was determined using two-way ANOVA: **p* < 0.05 indicates a significant result; ***p* < 0.01 a very significant result; ****p* < 0.001 a highly significant result; and *****p* < 0.0001 an extremely significant result.

**Figure 6 polymers-13-01883-f006:**
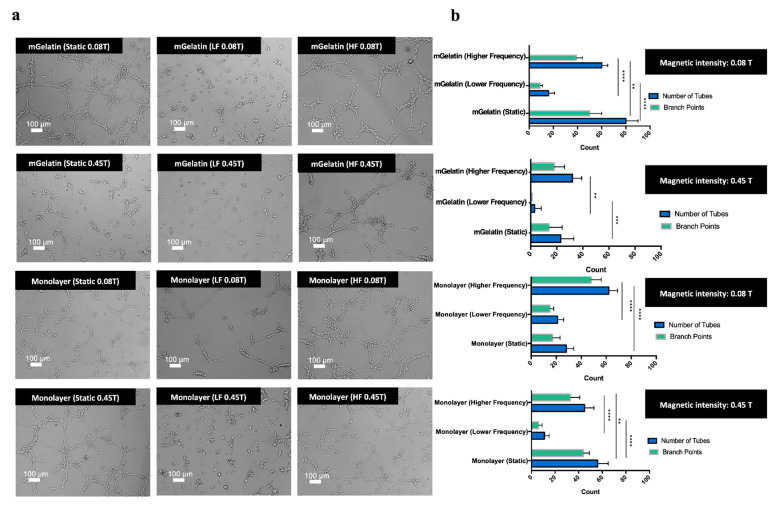
(**a**) Representative images of HUVEC sprouting obtained from MSCs conditioned media cultured in each condition (mGelatin vs monolayer; 0.08 T vs 0.45 T; static vs lower frequency vs higher frequency magnetic field). Scale bar: 100 µm. (**b**) Number of tubes and number of branch points measured for each analysis condition of the tube formation assay. (**c**) Representative images of the tube formation controls and respective measurement of tubes number and branch points. Endothelial cell culture media, rich in growth factor supplements, was used as the positive control. MSCs secretome, from MSCs cultured in polystyrene culture plate without magnetic exposure, was used as the negative control. Scale bar: 100 µm. All data were obtained from a pool of three independent MSC donors (A, B, C) and presented as means ± SD. Statistical significance was determined using two-way ANOVA: ***p* < 0.01 a very significant result, ****p* < 0.001 a highly significant result; and *****p* < 0.0001 an extremely significant result.

## Data Availability

Data available on request.

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
