# Peer review of "Magnetic Field Dynamic Strategies for the Improved Control of the Angiogenic Effect of Mesenchymal Stromal Cells"

_polymers, 2021, doi:10.3390/polym13111883_

Round 1
Reviewer 1 Report
Comments:
In this study, the authors showed the possibility to control VEGF-A secretion by MSCs through modulation of the magnetic field, offering attractive perspectives of a non-invasive therapeutic option for several diseases by revascularizing damaged tissues or inhibiting metastasis formation during cancer progression. It is an interesting study with a novel approach. The manuscript is very well written, and it has a nice flow of information. It is very easy to read and understand, the results were clearly presented. The scientific merits of this manuscript are high.
- In figure 6 authors can increase the resolution of the figure with high quality
- VEGF-A is the important indicator of angiogenesis, however, the author did not link the other proteins which are involved in the angiogenesis pathway.
- In the future I recommend to authors, they can study in-depth molecular mechanisms of the outcomes, they obtained in this study.
Author Response
Answers to the comments of Reviewer 1:
The authors would like to thank the Reviewer for the positive feedback and useful comments and suggestions. The paper has been revised according to the suggestions provided by the Reviewer and we believe its quality has been considerably improved. We hope these changes are satisfactory. All the changes introduced in the revised version of the manuscript are highlighted in yellow for a better perception of the Reviewers and the Editor.
Comments:
In this study, the authors showed the possibility to control VEGF-A secretion by MSCs through modulation of the magnetic field, offering attractive perspectives of a non-invasive therapeutic option for several diseases by revascularizing damaged tissues or inhibiting metastasis formation during cancer progression. It is an interesting study with a novel approach. The manuscript is very well written, and it has a nice flow of information. It is very easy to read and understand, the results were clearly presented. The scientific merits of this manuscript are high.
Comment 1: In figure 6 authors can increase the resolution of the figure with high quality.
The authors appreciate the comment and have accordingly increased the resolution of Figure 6 for a better understanding of the results for the readers. We also decided to complement the figure with images of HUVECs marked with CD31, used primarily to demonstrate the presence of endothelial cells. Please consider the improved Figure 6 on page 12, as well as the corresponding changes in the experimental section (page 6 line 211) and in the Results section (page 13 line 444).
“2.5. CD31 Immunostaining.
HUVECs immunocharacterization was carried out using CD31 antibody after cells fixation with 4% paraformaldehyde and blocking with 10% FBS (Fetal Bovine Serum, Gibco) in PBS solution. Primary antibody CD31 (1:50 dilution, mouse antibody, Dako) in block solution was incubated overnight. Secondary antibody Alexa 546 (1:500 dilution, goat anti-mouse, Abcam) was added and incubated for 30 min. Images were obtained using a fluorescence microscope (LEICA DM IL LED with EC3 camera system) and a confocal laser scanning microscope (LSM 700/ Carl Zeiss).”
“A monolayer of HUVECs culture was marked with CD31 antibody (specific endothelial cell marker) to confirm the endothelial lineage of HUVECs and understand their ability for sprouting and tube formation (Figure 6c).”
Comment 2: VEGF-A is the important indicator of angiogenesis, however, the author did not link the other proteins which are involved in the angiogenesis pathway.
The authors acknowledge the suggestion of the Reviewer and have decided to expand the paragraph about angiogenesis, by adding the text shown below, in the Introduction section in page 1 and lines 34 of the revised version of the manuscript to indicate other fundamental molecules involved in the angiogenic pathway, including fibroblast growth factors (FGFs), platelet-derived growth factor (PDGF), hepatocyte growth factor (hGF) and angiopoietins (ANGs). With this information, we hope to provide a more complete introduction regarding the angiogenesis phenomena.
“Briefly, the complex mechanism of angiogenesis is initiated when the increase in concentration of pro-angiogenic factors - such as vascular endothelial growth factor (VEGF), angiopoietins, fibroblast growth factors (bFGFs), hepatocyte growth factors (hGFs), platelet-derived growth factor (PDGF), among others - produced by inflammatory or tumor cells activate quiescent endothelial cells from an existing vessel in response to injury and/or hypoxia. The activated cells differentiate into tip cells, creating elongated sprouts/new vessels toward the stimulus through active migration. After capillary formation, endothelial cells secrete attractant molecules to recruit perivascular cells, in the capillaries, or smooth muscle cells, in larger vessels.”
Additionally, we added RT-PCR results shown below and added the Supplementary Data as Figure 2S, concerning the MSCs expression of other angiogenic genes such as HGF and FGF under static magnetic field, the configuration that provided better proangiogenic results, alongside with high frequency magnetic dynamic mode. These results demonstrated that the magnetic field does not seem to impact on the expression of HGF and FGFs in the same way that it does with VEGF, which was why in this work we focused on the possibility of magnetically modulate the secretion of VEGF from MSCs. More studies on the effect of static magnetic field on the behavior of mesenchymal stromal cells (MSCs) are part of another publication, which was just accepted for publication and cited in the revised version of this manuscript as [37]. Changes in the manuscript are present in page 10 line 331.
“Other angiogenic genes were also analyzed (bFGF and hGF) alongside VEGF-A for the static magnetic regime under 0.08 T and compared with the respective condition not exposed to magnetic stimulation – 0 T (Figure S2). However, the magnetic field does not seem to have an impact on the expression of bFGF and hGF angiogenic genes by MSCs, revealing no differences between the conditions 0.08 T and 0 T. For this reason, our study was mainly focused on the effect of the magnetic field on the expression and secretion of VEGF-A.”
Comment 3: In the future I recommend to authors, they can study in-depth molecular mechanisms of the outcomes, they obtained in this study.
The authors are in total agreement with this comment and welcome the suggestion of the Reviewer. We are currently preparing more studies in this topic involving western blot analysis of different molecules, for assessing the expression of early differentiation genes by MSCs under magnetic exposure. Proteomic studies and RNA sequencing are also planned for the future studies to explore other potential effects that the magnetic field might be causing on the MSCs.

Reviewer 2 Report
The manuscript entitled 'Magnetic field dynamic strategies for improved control of the angiogenic effect of mesenchymal stromal cells' by Mantua et al. focussed on angiogenic effects of magnetic stimulation on endothelial cells by modulation of VEGF-A segregation by MSCs.
The study is a very interesting study. It is of importance as an improved understanding of these effects can results in novel insights into blood vessel growth in health and disease.
The manuscript is well written and has great figures. Furthermore, it addresses most aspects which are relevant in this context.
However, I would like to ask the authors to comment on the effect of VEGF-A mediated stimulation of MSC on lymphatic endothelial cells as the lymphatic vasculature plays also important roles in regenerative medicine and human disease. What about a potential impact of magnetic field on lymphangiogenesis - especially in for patients with lymphatic dysfunction after cancer treatment.
Author Response
Answers to the comments of Reviewer 2:
The authors would like to thank the Reviewer for the positive feedback and interesting question. The changes introduced in the revised version of the manuscript are highlighted in yellow for a better perception of the Reviewer and the Editor.
Comments:
The manuscript entitled 'Magnetic field dynamic strategies for improved control of the angiogenic effect of mesenchymal stromal cells' by Mantua et al. focused on angiogenic effects of magnetic stimulation on endothelial cells by modulation of VEGF-A segregation by MSCs. The study is a very interesting study. It is of importance as an improved understanding of these effects can results in novel insights into blood vessel growth in health and disease.The manuscript is well written and has great figures. Furthermore, it addresses most aspects which are relevant in this context.
Comment 1: However, I would like to ask the authors to comment on the effect of VEGF-A mediated stimulation of MSC on lymphatic endothelial cells as the lymphatic vasculature plays also important roles in regenerative medicine and human disease. What about a potential impact of magnetic field on lymphangiogenesis - especially in for patients with lymphatic dysfunction after cancer treatment.
The authors would like to thank the Reviewer for this comment and, although we have no experience with lymphatic endothelial cells, we would like to address this comment with a hypothetical theory. Lymphangiogenesis consists in the formation of lymphatic vessels from pre-existing vessels, in an apparently similar mechanism as blood vessel development in angiogenesis, while being responsible for an important physiological role in homeostasis, metabolism and immunity. Additionally, lymphatic endothelial cells possess specific markers, such as VEGF-C and VEGFR-3, and pro-lymphangiogenesis inducers include VEGF-C, hyaluronic acid and ephrin-B2 molecules. Our hypothesis is that the if the magnetic field impacts on the VEGF pathway, promoting the overexpression of VEGF-A by the mesenchymal stromal cells (MSCs) as shown in the present manuscript, it is plausible to consider that it might also impact on the expression of other proteins of the VEGF sub-family.
In that case, it would be expected that an enriched conditioned media containing oversecreted VEGF-C proteins, due to magnetic stimulation of MSCs, would induce the formation of the Turing patterns correlated with the interaction between VEGF-C and lymphatic endothelial cells. Therefore, the magnetic stimulation may contribute to positive or negative regulation of lymphangiogenesis. If this is verified, it would definitely be impactful for patients with lymphatic dysfunction as it would provide a therapeutic option, in particular, for cancer patients. Nonetheless, the closest structural and functional relative of VEGF-C is reported to be VEGF-D, and instead our findings have been focused on VEGF-A. We have never studied the expression of VEGF-C, consequently this answer can only be interpreted as a hypothetical potential for the use of magnetic stimulation on lymphangiogenesis phenomena. We considered this suggestion of the Reviewer very interesting and decided to include this possibility in the Conclusions section of the manuscript as potential future work (page 17, line 633).
“Exploring the impact of the magnetic field over the secretion of other molecules by MSCs might provide valuable information and novel therapeutic routes for several diseases. In particular, the potential impact of the magnetic stimulation on other molecules from the VEGF family, such as VEGF-C and VEGF-D, might also offer perspectives on the application of this approach to promote lymphangiogenesis in lymphatic conditions.”

Reviewer 3 Report
Manjua et al. show an experimental study on the possible role that electromagnetic fields have on the expression of markers associated with angiogenesis.
-The introduction presented by the authors has serious deficiencies from the biological and mechanistic point of view. As a whole, HUVECs cells are more specific and possess a lineage capacity towards the angiogenic line. In this regard, the authors should remember the influence of Wharton's jelly. Basing and justifying the angiogenic role only on VEGF is absolutely wrong and does not allow to create a study hypothesis.
-The methodology is incomplete. What is the cellular origin?
-Explain the In vitro endothelial cell tube formation assay better.
-Justify why you use an ANOVA between two variables? Why don't you use IQR?
-In vitro studies are very limited and have serious deficiencies. In the first place, studies are necessary where the cell viability is specifically seen, as well as the phase of the cell cycle where the cells are found.
-It is necessary to study specifically the effects on the apoptosis process, to know what the effect of electromagnetic fields is. Authors should add quantifications of BAX / BCL-2, PARP-1 levels. TUNEL technique, among others.
-The angiogenesis markers must be amplified, it must be specifically studied that the cells are differentiating towards that lineage. Authors should include CD31, CD34, PAI-I.
-It is very necessary to add eNOS and iNOS to verify this angiogenic effect.
-Figure 6 is very deficient, it does not allow us to observe the differences shown in the attached graphs.
- The discussion follows a primitive argument similar to the introduction. But the conclusions that are tried to obtain are too categorical. Authors must be more explicit and assertive.
-Can the authors explain the statement of line 470?
-The manuscript has numerous grammatical errors. Authors should improve their expressions in English grammar.
Author Response
Answers to the comments of Reviewer 3:
The authors would like to thank the Reviewer for the constructive feedback and useful comments and suggestions. The paper has been revised according to the suggestions provided by the Reviewer and we believe its quality has been considerably improved. We hope these changes are satisfactory. All the changes introduced in the revised version of the manuscript are highlighted in yellow for a better perception of the Reviewers and the Editor.
Comments:
Manjua et al. show an experimental study on the possible role that electromagnetic fields have on the expression of markers associated with angiogenesis.
Comment 1: The introduction presented by the authors has serious deficiencies from the biological and mechanistic point of view. As a whole, HUVECs cells are more specific and possess a lineage capacity towards the angiogenic line. In this regard, the authors should remember the influence of Wharton's jelly. Basing and justifying the angiogenic role only on VEGF is absolutely wrong and does not allow to create a study hypothesis.
The authors understand the concern of the Reviewer regarding our focus on VEGF-A to justify the pro-angiogenic or anti-angiogenic behavior of MSCs. In particular, we agree that in the Introduction it might have been misleading to the readers to mention only VEGF and therefore we have added a new paragraph in this section (page 1 line 34) to briefly explain the mechanisms involved in the angiogenesis phenomena.
“Briefly, the complex mechanism of angiogenesis is initiated when the increase in concentration of pro-angiogenic factors - such as vascular endothelial growth factor (VEGF), angiopoietins, basic fibroblast growth factors (bFGFs), hepatocyte growth factors (hGFs), platelet-derived growth factor (PDGF), among others - produced by inflammatory or tumor cells activate quiescent endothelial cells from an existing vessel in response to injury and/or hypoxia. The activated cells differentiate into tip cells, creating elongated sprouts/new vessels toward the stimulus through active migration. After capillary formation, endothelial cells secrete attractant molecules to recruit perivascular cells, in the capillaries, or smooth muscle cells, in larger vessels.”
Nonetheless, in this work we have only performed in vitro studies, which are very limited in comparison with the complexity of the in vivomicroenvironment, where the effect of Wharton’s jelly tissue would definitely be a relevant factor, given how the cells in Wharton’s jelly express several stem cell genes. Also, the interaction with other cell types and recruiting molecules in the in vivo microenvironment is fundamental for the angiogenesis process, which is very difficult to be reproduced in in vitro studies. The main goal of this work was to investigate the possibility to magnetically modulate sprouting and tube formation behavior of endothelial cells. This was actually achieved in an indirect way, i.e. by using the conditioned media of MSCs that had been previously magnetically stimulated. A clear relation between endothelial cell sprouting and tube formation and the VEGF-A expression by MSCs was found. Although VEGF-A is one of the most important angiogenic molecules, we are aware that many other angiogenic proteins might also be simultaneously involved in the process and might be affected by the magnetic stimulation. For this reason, in the first step of our work we have analyzed the potential impact of static magnetic field with an intensity of 0.08 T also on hGF (hepatocyte growth factor) and bFGF (basic fibroblast growth factor). We totally agree with the Reviewer with the need to clarify these aspects in the present manuscript and for this reason, we opted to include the RT-PCR results obtained regarding the magnetic effect on the expression of other angiogenic molecules such as hGF and bFGF in the presence and absence of a static magnetic field of 0.08 T, in Figure S2 of Supplementary Data. The results showed that these molecules were not as affected by the magnetic field as VEGF-A, which motivated us to proceed our research work with a focus on VEGF-A. Changes in the manuscript are present in page 10 line 331 as follows:
“Other angiogenic genes were also analyzed (bFGF and hGF) alongside VEGF-A for the static magnetic regime under 0.08 T and compared with the respective condition not exposed to magnetic stimulation – 0 T (Figure S2). However, the magnetic field does not seem to have an impact on the expression of bFGF and hGF angiogenic genes by MSCs, revealing no differences between the conditions 0.08 T and 0 T. For this reason, our study is mainly focused on the effect of the magnetic field on the expression and secretion of VEGF-A.”
Despite the clear relation between HUVEC angiogenesis and the capacity to up and down regulate the VEGF-A expression by MSCs shown in this work, it does not exclude however the need to evaluate other cell mechanisms which may also contribute for angiogenesis. In this regard, further studies are currently on going in this topic involving western blot analysis of different molecules, for assessing the expression of early differentiation genes by MSCs under magnetic exposure. Proteomic studies and RNA sequencing are also planned in near future to investigate other potential effects that the magnetic field might be causing on the MSCs. We hope to be able to consider these additional results as topic of a future publication.
Comment 2: The methodology is incomplete. What is the cellular origin?
The authors acknowledge the Reviewer for the careful reading and for noticing this incomplete information. In this work we used mesenchymal stromal cells from the human bone marrow of different healthy donors, which provided informed consent for our use within our Stem Cell Engineering research group. These bone marrow samples were collected from an oncology institute in Portugal under a collaboration with our group. The samples were processed, and the mesenchymal stromal cells were collected within our group for cell culture experiments. The HUVECs cell line we used in this study is commercially available and was purchased from Lonza and expanded within our group for cellular experiments. We further clarified this information by including the text below in revised version of the manuscript, in the experimental section 2.3. Cell culture, page 5 line 174.
“2.3. Cell culture.
MSC lines used in this paper derive from human bone-marrow samples, donated by healthy donors under informed consent, accordingly with the Directive 2004/23/EC of the European Parliament and of the Council of 31 March 2004 on setting standards of quality and safety for the donation, procurement, testing, processing, preservation, storage, and distribution of human tissues and cells (Portuguese Law 22/2007, June 29), with the approval of the Ethics Committee of the respective clinical institutions[36]. These cells belong to the cell bank available at Stem Cell Engineering Research Group (SCERG), iBB-Institute for Bioengineering and Biosciences at Instituto Superior Técnico (IST). Bone marrow samples were retrieved from Instituto Português de Oncologia Francisco Gentil, Lisboa-Portugal under collaboration with iBB-IST. Human samples were obtained from healthy donors Isolated cells were cryopreserved in liquid/vapor nitrogen tanks. Isolated human bone marrow MSCs (BM MSCs) were cultured on low-glucose Dulbecco’s Modified Eagle Medium (DMEM, Gibco Grand Island NY) supplemented with 10% fetal bovine serum (FBS MSC qualified, Gibco) and 1% antibiotic-antimycotic (Gibco) and kept at 37ºC, 5% CO2 and 21% O2 in a humidified atmosphere. The phenotype of MSCs under magnetic exposure was confirmed in our previous study [37]. Briefly, the cells were tested by flow cytometry after 5 days, with and without magnetic exposure, for expression of cell surface markers indicative of MSC using a panel of mouse anti-human monoclonal antibodies (PE-conjugated) against: CD73+, CD90+, CD105+, CD14-, and human leukocyte antigen HLA-DR- (all from Biolegend).Three independent donors were used on the experiments (n=3, donor A, B, C). HUVECs used in this report are commercially available and were purchased from Lonza (Basel, Switzerland), and expanded in commercial endothelial growth medium-2 (EGM-2, Lonza) and kept at 37ºC, 5% CO2 in a humidified atmosphere. Medium renewal was performed every 3-4 days. All the cellular experiments were performed between passages 4 and 7.”
Comment 3: Explain the In vitro endothelial cell tube formation assay better.
The authors acknowledge the comment of the Reviewer, which gives us the possibility to better describe the experiment in the revised version of the manuscript. The effect of the magnetic field on the endothelial cells were accessed through an indirect method. As such, a culture of MSCs, supplemented with basal media to prevent the presence of VEGF, was first exposed to different magnetic field regimes for 24h. The supernatant of the MSCs was collected at the end of the experience and appropriately stored. HUVECs were then seeded on top of Matrigel coated plates, a condition that was preferred to potentiate the maturation of the endothelial cells, and the HUVECs were supplemented with the supernatant of MSCs and allowed for 6 hours to observe the formation of tube-like structures. HUVECs supplemented with completed endothelial cell media, containing regular concentration of VEGF, and HUVECs supplemented with the supernatant from MSCs non exposed to magnetic stimulation in the same basal media which do not contain VEGF, were used as positive and negative controls, respectively. The comparison between positive (to induce the maturation of HUVECs into tube structures) and negative control (where no tubes were expected) with the conditions using the supernatant from MSCs exposed to magnetic stimulation allowed to observe the morphological differences in HUVECs regarding the formation of new tubes and classify the more proangiogenic and anti-angiogenic conditions. A more clear and complete description of the tube formation methodology was included in the Experimental section of the revised version of the manuscript in page 7 line 257.
"2.9. In vitro endothelial cell tube formation assay.
To evaluate the effect of exogenous VEGF-A supplementation on angiogenic properties, a three-dimensional capillary-like tube formation assay was performed. For this experiment MSCs were incubated with Endothelial Cell Basal Medium (EBM-2, Lonza), VEGF free media, during magnetic application. Simultaneously, HUVECs (2x104 cells) were cultured on Matrigel (50 µl/well) in a 96-well plate. The conditioned media (without VEGF and FBS supplements) obtained from MSCs, cultured on scaffolds or in monolayer under magnetic effect, was added to each well. The observation of the induced HUVECs sprouting on the conditions of analysis was compared with positive and negative controls for this experiment, respectively controls where HUVECs were supplemented with Endothelial Cell Growth Media-2 (EGM-2), rich in growth factor supplements to promote HUVECs sprouting, and the secretome from MSCs growing in polystyrene without magnetic exposure in order to assess the effect of the residual secretion of growth factors by MSCs that were never exposed to magnetic stimulation. After incubation for 6h at 37ºC, three images of the center of each well were taken using a light microscope (Leica DM IL LED with EC3 camera system). The number of tubular-like structures and branch points formed were counted using ImageJ (NIH) software. All conditions were tested in triplicates using three different donors (donors A, B, C)."
Comment 4: Justify why you use an ANOVA between two variables? Why don't you use IQR?
The goal of statistical analysis was to compare different conditions in relation with the MSCs cultured in monolayer and gelatin obtained from three independent donors (n=3). The statistical analysis was performed with Prism GraphPad tool, using the analysis of variance (two-way ANOVA) which was the statistical test recommended for the number of variables and biological replicates considered. Two-way ANOVA was used to analyze the difference between the means of more than two groups when the goal is to know how two independent variables (gelatin and monolayer), in combination, can affect a dependent variable (magnetic regime). On the other hand, the interquartile range (IQR) is particularly advantageous when there are outliers that disproportionately affect the range of the data set. With IQR, the middle 50% of the data set will not be affected by the outliers. Although we agree that IQR would be a valuable test considering the variability within the MSC donors, we also believe two-way ANOVA was an appropriate statistical test for the data set we obtained without losing any information.
Comment 5: In vitro studies are very limited and have serious deficiencies. In the first place, studies are necessary where the cell viability is specifically seen, as well as the phase of the cell cycle where the cells are found.
The authors agree with the comment of the reviewer regarding the need for a better characterization of the mesenchymal stromal cells (MSCs) and Human Umbilical Vein Endothelial Cells (HUVECs). Regarding MSCs, we performed live/dead assay to identify the toxicity of the magnetic field conditions. The images obtained from the live/dead assays, shown in Figure 3a in the revised version of the manuscript, confirmed a high viability of the cells in the presence of the magnetic field, based on the reduced number of death cells observed. It is true that we did not perform any study to verify the phase of cell cycle of the MSCs during the magnetic exposure. Nevertheless, the phenotypic characterization of MSCs under magnetic exposure by flow cytometry was performed being part of a previous work which was just accepted for publication and is referenced as [37] in the revised version of this manuscript. The flow cytometry studies confirmed that MSCs maintain their phenotype under magnetic exposure. This information was added in the revised version of the manuscript page 5, line 188.
“The phenotype of MSCs under magnetic exposure was confirmed in our previous study [37]. Briefly, the cells were tested by flow cytometry after 5 days, with and without magnetic exposure, for expression of cell surface markers indicative of MSC using a panel of mouse anti-human monoclonal antibodies (PE-conjugated) against: CD73+, CD90+, CD105+, CD14-, and human leukocyte antigen HLA-DR- (all from Biolegend).”
Calcein staining assays were also performed to ensure HUVECs survival. A representative image was included as Figure S3, in the Supplementary Data, as a representative image of HUVECs in monolayer and during tube formation. The phase of cycle of HUVECs during tube formation was not investigated, however morphological differences between a cell culture of HUVECs, related with branching formation, before and after tube formation assay were clearly observed. This information was included in the revised version of the manuscript, page 13, line 447.
“Cell viability of HUVECs, prior tube formation assay in a monolayer culture, and during tube formation functional assay was also assessed using Calcein to mark living cells and evaluate cell survival during the process of vessel branching (Figure S3).”
Comment 6: It is necessary to study specifically the effects on the apoptosis process, to know what the effect of electromagnetic fields is. Authors should add quantifications of BAX / BCL-2, PARP-1 levels. TUNEL technique, among others.
We agreed with the comment of the Reviewer that quantifications of the apoptotic process of MSCs during exposure to the magnetic field, including information of DNA fragmentation and cell cycle, with quantification of BAX/ BCL-2, PARP-1 levels, would be very interesting in this study and could further complement the manuscript information. However, the main focus of this manuscript was to prove the ability to magnetically modulate the angiogenic phenomena while selecting the best magnetic field conditions which may lead to the desirable result, i.e. promote or inhibit angiogenesis, according to the applications. Still, as mentioned in the manuscript, further studies are required to clarify the cell mechanisms which are affected by the magnetic field and thus influencing VEGF-A expression. Based on the proliferation studies included in this work, it seems clear that cell apoptosis should assume a relevant contribution to the VEGF-A production. However, we are aware that it might be other mechanisms with significant contribution. For this reason, the authors are currently developing studies aiming at assessing the influence of magnetic field in the expression of early differentiation genes for MSCs. Also, proteomic studies and RNA sequencing will be developed in near future in order to clarify other potential effects and cell mechanisms of MSCs affected by the magnetic field and that may allow a deeper understanding of the MSC magnetic stimulation of the HUVEC angiogenesis. We hope that these studies may be considered as the topic of a future publication.
Comment 7: The angiogenesis markers must be amplified, it must be specifically studied that the cells are differentiating towards that lineage. Authors should include CD31, CD34, PAI-I. It is very necessary to add eNOS and iNOS to verify this angiogenic effect.
The authors agree that HUVECs could have been further characterized during sprouting to assess cell differentiation. We have actually done some of the analysis suggested by the Reviewer in order to characterize the endothelial lineage of the cells. We have analyzed CD31 specific marker for endothelial cells by HUVECs. Images of the results were included as Figure 6 in the revised version of the manuscript and confirmed the endothelial lineage of the cells used for tube formation analysis. This additional information was added in the revised version of the manuscript in page 12 and in the Results section in page 13 line 444.
“A monolayer of HUVECs was marked with CD31 antibody (specific endothelial cell marker) to confirm the endothelial lineage of HUVECs and understand their ability for sprouting and tube formation (Figure 6c).”
We also acknowledge the interest on staining the cells during tube formation with CD34, as a marker for endothelial cell differentiation and endothelial progenitor cells, and even PA-I, as a marker involved in the haemostatic balance of the cells. In the same way, we also believe that the quantification of eNOS, endothelial nitric oxide synthase associated with inflammatory processes in cardiovascular diseases and crucial in the regulation of the vascular endothelium, and iNOS, inducible nitric oxide synthases which is one of the reactive oxygen and nitrogen metabolite-metabolizing enzymes, would be very interesting and impactful for our study. Nonetheless, these studies are already planned and will be performed in near future as a way to complement our work in this field and as part of a future publication on the effect of the magnetic field at the intracellular and molecular level.
Comment 8: Figure 6 is very deficient, it does not allow us to observe the differences shown in the attached graphs.
The authors agree with the comment of the Reviewer. Figure 6 was improved with better image resolution and contrast for a better understanding of the readers. Additionally, the authors have also added complementary images of HUVECs monolayer before the in vitro tube formation functional assay. In this way, it is possible to morphologically compare HUVECs in cell culture and observe their maturation during sprouting phenomena within tube formation assay. We also decided to add an image of HUVECs marked with CD31 endothelial specific antibody to confirm the endothelial phenotype of the cells. The revised Figure can be found in page 12 of the manuscript.
Comment 9: The discussion follows a primitive argument similar to the introduction. But the conclusions that are tried to obtain are too categorical. Authors must be more explicit and assertive.
The authors accept this comment and have modified the conclusions section so that the information can flow in a more assertive style with an explicit explanation of the most relevant information obtained within this study. The Conclusions section was modified (page 17) as follows:
“These results prove that magnetic stimulation may either function as a tool to reduce tumor growth in association with cancer treatments (using LF operation modes) or a possibility to promote the regeneration of diseased blood vessels (static and HF modes) in a contribution for vascular diseases treatments through cellular therapies.
Complementary studies are yet required for a deeper understanding of the cell mechanisms able to further investigate the magnetic dependence of angiogenic VEGF-A expression by MSCs, such as RNA sequencing, proteomic studies and western blots to identify the activated pathways in MSCs during magnetic exposure. Exploring the impact of the magnetic field over the secretion of other molecules by MSCs might provide valuable information and novel therapeutic routes for several diseases. In particular, the potential impact of the magnetic stimulation on other molecules from the VEGF family, such as VEGF-C and VEGF-D, might also offer perspectives on the application of this approach to promote lymphangiogenesis in lymphatic conditions. Nonetheless, the present investigation reveals a modulatory effect of magnetic field regimes on MSCs behaviour, thus offering the possibility to regulate angiogenesis phenomena.”
Comment 10: Can the authors explain the statement of line 470?
“Similarly, anti-angiogenic effect induced by LF magnetic field resulted in lower tube formation and vessel branching than that obtained for HUVECscultured in TCP cell plates without magnetic field stimuli (negative control). These results prove that the magnetic field may effectively regulate angiogenesis, promoting a pro-angiogenesis and anti-angiogenesis effect depending only on a proper selection of the magnetic field parameters, i.e. magnetic field intensity and/or magnetic field variation dynamics.”
The authors totally understand the confusion and acknowledge the Reviewer for noticing the mistake in the sentence, helping us to improve the quality and the perception of the results described in this manuscript The error has been corrected in the revised version of the manuscript (page 14 line 505) as follows:
“Similarly, anti-angiogenic effect induced by LF magnetic field resulted in lower tube formation and vessel branching than that obtained for MSCscultured in TCP cell plates without magnetic field stimuli (negative control). These results prove that the magnetic field may effectively regulate angiogenesis, promoting a pro-angiogenesis and anti-angiogenesis effect depending only on a proper selection of the magnetic field parameters, i.e. magnetic field intensity and/or magnetic field variation dynamics.”
With this statement the authors are acknowledging the morphological differences observed on HUVECs during tube formation assay and attributing these effects to the conditioned media from MSCs exposed to different magnetic field regimes. The comparative analysis included in the manuscript was established considering as negative control the least angiogenic condition, i.e. a non-supplemented HUVEC cultured in condition media obtained from MSCs non-stimulated by the magnetic field, where the release of growth factors from MSCs was not associated with the magnetic field effect but it was simply due to the MSCs metabolic pathways, and a positive control as the regular conditions for tube formation, with HUVECs supplemented with VEGF media and Matrigel. Then it was possible to conclude that magnetic field allowed for up and down regulation of angiogenesis, which could be prompted by simple adjustment of the magnetic field condition, i.e. an anti-angiogenic condition associated to LF magnetic field, where the tube formation was lower than that obtained in the negative control (anti-angiogenic effect) and a pro-angiogenic condition, with higher tube formation and sprouting obtained at static and HF magnetic fields, which in some circumstances (static magnetic fields) slightly surpassed the number of tubes obtained in the positive control.
Comment 11: The manuscript has numerous grammatical errors. Authors should improve their expressions in English grammar.
The authors have taken into consideration the Reviewer suggestion and have accordingly revised carefully the manuscript and the writing quality was improved.

Reviewer 4 Report
The work is focused on the analysis of mesenchymal stem cell behaviour and angiogenic output, when grown on iron doped gelatin layers and exposed to magnetic fields with different intensity and frequencies. The paper explores the effects of magnetic fields with graded intensity and dynamic regimes on VEGF-A transcription and secretion, ultimately controlling the endothelial cell potential in an in vitro angiogenesis model. The results document a differential behaviour in relation tot the intensity and frequency of magnetic field, suggesting a potential modulating control of angiogenesis both in positive and negative application regimes.
The paper in the whole is well designed and results sound.
Comments:
- At page 3, please indicate the real concentration of gelatin within the biomaterial, if 8% w/w or 8% m/v.
- Do the authors normalize VEGF-A protein quantification on the number of cells or protein content for each experimental condition? As it is, the numbers may derive form the final number of cells remaining on the biomaterial after the exposure to magnetic stimulation and not the ability of the cell population to metabolically produce the angiogenic factor. Please consider this type of data elaboration to formulate and discuss the results.
- In Matrigel in vitro formation assay, there is no indication about the sampling of MSC conditioned medium, if from as single donor, and what/why, or as a pool of the 3 donors. This information should be added.
Author Response
Answers to the comments of Reviewer 4:
The authors would like to thank the Reviewer for the positive feedback and useful comments and suggestions. The paper has been revised according to the suggestions provided by the Reviewer and we believe its quality has been considerably improved. We hope these changes are satisfactory. All the changes introduced in the revised version of the manuscript are highlighted in yellow for a better perception of the Reviewers and the Editor.
Comments:
The work is focused on the analysis of mesenchymal stem cell behaviour and angiogenic output, when grown on iron doped gelatin layers and exposed to magnetic fields with different intensity and frequencies. The paper explores the effects of magnetic fields with graded intensity and dynamic regimes on VEGF-A transcription and secretion, ultimately controlling the endothelial cell potential in an in vitro angiogenesis model. The results document a differential behavior in relation to the intensity and frequency of magnetic field, suggesting a potential modulating control of angiogenesis both in positive and negative application regimes. The paper in the whole is well designed and results sound.
Comment 1: At page 3, please indicate the real concentration of gelatin within the biomaterial, if 8% w/w or 8% m/v.
We thank the Reviewer for bringing this to our attention and we would like to confirm that the concentration of the gelatin is 8% m/v. The manuscript has been appropriately revised in this section. We thank the reviewer for bringing this to our attention and we would like to confirm that the concentration of the gelatin is 8% m/v. The manuscript has been appropriately revised and this information was included in the section of Materials and Methods in page 4, line 131 of the revised version of the manuscript.
“Porcine skin gelatin (8% m/v, type A, G2500, Sigma-Aldrich) was dissolved in milli-Q water at 60ºC. MNPs were dispersed by sonication in the polymeric aqueous solution.”
Comment 2: Do the authors normalize VEGF-A protein quantification on the number of cells or protein content for each experimental condition? As it is, the numbers may derive form the final number of cells remaining on the biomaterial after the exposure to magnetic stimulation and not the ability of the cell population to metabolically produce the angiogenic factor. Please consider this type of data elaboration to formulate and discuss the results.
The authors acknowledge the Reviewer for this pertinent comment. In this study we did not normalize the protein quantification on the number of cells or the protein content. We assumed the toxicity associated with the magnetic field effect as a key factor, which was at least partially responsible for the changes in the secretion of VEGF by MSCs. However, it is also possible that other metabolic changes, aside from cell death, are associated with the magnetic field influence on the cells and may also impact on the secretion of VEGF-A. We are currently developing further studies aiming to deeply understand the regulatory mechanisms involved in the magnetic stimulation of MSCs and the suggestion of the Reviewer to normalize the VEGF-A secretion values with the number of cells quantified via AlamarBlue will certainly be considered as a way to conclude about the contribution of other metabolic pathways on the regulation of VEGF production by MSCs.
Comment 3: In Matrigel in vitro formation assay, there is no indication about the sampling of MSC conditioned medium, if from as single donor, and what/why, or as a pool of the 3 donors. This information should be added.
The authors acknowledge and thank the Reviewer from bringing this to our attention. The tube formation assay was performed using three independent donors (A, B, C) as the rest of the experiments. However, we decided to show the results in Figure 6 as a pool of the 3 donors. We believed that presenting those results with individualized donors would be too confusing for the readers and would make the main message of the study more difficult to understand. Nonetheless, the Reviewer is completely right about the need in adding this information. In this sense, the caption of Figure 6 (page 12 of the manuscript) was revised accordingly.

Round 2
Reviewer 3 Report
The authors have carried out a revision of their manuscript that is not complete with the specified indications. I would ask the authors to answer the questions precisely and specifically make the changes. Authors respond extensively in report but in a cumbersome way to make the reviewer get lost. The authors should review all the previous points, but I would insist on several points:
-The authors have not made a modification of the introduction with the comments. It is too long and the translational importance of the study is not specified.
- The material and methods section has been improved. However, there are points that are not clear. Cell typing? Time? The typing of cells?
-The point 2.4 should be adequately explained. Please expand this point appropriately. Did the authors use fluorescence typing?
-The authors must provide in detail all the data of their providers. This information is not specified in the materials and methods section.
-How do the authors justify the sample size for the publication to have a translational character?
-The description of the results has not improved. The figures are still not very explanatory, for example figure 4 is not suitable among others.
-Pay the figure 6 is not valuable. The reader cannot see anything. Manification is not described. Please, the authors should enlarge it or split the figures. Authors must put readings on all panels.
-The authors must unify their figures and the way in which they present the results. This oil is critical and is currently not assessable.
- The data and p values must be specified at all times.
-The authors must specify the methodology of analysis and interpretation.
-How do they do and justify cell viability? Cycle? Why is flow cytometry not done specifically?
-The discussion is absolutely inappropriate. The authors have not remodeled anything. The discussion must have a translational character and have a more modern bibliography.
-Explain the supplementary material adequately. Authors must include all unmounted images as well (fluorescence, etc .....).
-The authors must include a graphic summary.
-The conclusion is too long and the authors extend themselves without giving the reader a clear and precise idea of the results.
-The authors have not checked the grammar of their manuscript. Authors should use the publisher's central services.
Author Response
Magnetic field dynamic strategies for improved control of the angiogenic effect of mesenchymal stromal cells
Ana C. Manjua, Joaquim M.S. Cabral, Frederico Castelo Ferreira*, Carla A. M. Portugal*
Manuscript ID: polymers-1171347
Answers to the comments of Reviewer 3:
The authors would like to thank the Reviewer for the constructive feedback and useful comments and suggestions. The authors did their best to deal conveniently with all the comments and criticisms raised by the Reviewer. The manuscript was revised according to the author’s best perception of the Reviewer’s requests, and we hope the changes included in the revised version of the manuscript may be satisfactory. All the changes introduced in the revised version of the manuscript are highlighted in yellow and blue color was selected to highlight linguist corrections for a better perception of the Reviewers and the Editor.
Comments:
The authors have carried out a revision of their manuscript that is not complete with the specified indications. I would ask the authors to answer the questions precisely and specifically make the changes. Authors respond extensively in report but in a cumbersome way to make the reviewer get lost. The authors should review all the previous points, but I would insist on several points:
The authors have taken into consideration the comment of the Reviewer and, as a result, some paragraphs of the introduction have been removed to clarify the main message of the manuscript. The translation importance of the study was specified in the following paragraph of page 3 line 98:
“As stated, the magnetic field can influence cellular processes, however the regulation of these effects remain underexplored. Hence, this study aims to provide a better understanding of the capacity to non-invasively modulate the angiogenesis process in endothelial cells via stimulation of VEGF-A production by MSCs, based on the variation of the magnetic field conditions, i.e. dynamic regimes and intensity. This study compares for the first time the impact of static and dynamic magnetic fields, applied at different intensities and variation regimes, on the secretion of angiogenic genes, mainly focusing on VEGF-A (key molecule in angiogenesis development) by MSCs cultured on magnetic-responsive gelatin scaffolds - mGelatin (Figure 1). Ultimately, this work envisages the use of MSCs as regulators/ stimulators of vessel sprouting in vitro and highlights the therapeutic clinical translation potential of the combination of MSCs and magnetic field. The capacity of the magnetic field to regulate angiogenic events may be regarded as an embryonary strategy that may prone to future development of magnetic-based therapies for the treatment of several vascular-related diseases. For instance, magnetic hydrogels loaded with MSCs may be injected in situ or used as an implantable stent to allow for the regeneration of damaged tissues and blood vessels through overexpression of proangiogenic genes under non-invasive external magnetic stimulation. Finally, the possibility to downregulate angiogenesis with magnetic stimuli can also be clinically translated as a powerful vascularization inhibition tool for tumor metastasis during cancer progression.”
The material and methods section has been improved. However, there are points that are not clear. Cell typing? Time? The typing of cells?
The authors have further expanded the Experimental section 2.3 according to their best perception of the information requested by the Reviewer. Additional information about cell cultures used and the MSCs donor condition, namely the gender and age of the donor and the isolation year was added in the revised version of the manuscript. All cellular experiments were performed with passages between 4 and 7. The modifications on this section are presented below and added in page 4 line 170 of the revised version of the manuscript. The authors hope to have satisfactorily addressed the comment of the Reviewer.
“2.3. Cell culture.
MSC lines used in this work derive from human bone-marrow samples, donated by healthy donors under informed consent, accordingly with the Directive 2004/23/EC of the European Parliament and of the Council of 31 March 2004 on setting standards of quality and safety for the donation, procurement, testing, processing, preservation, storage, and distribution of human tissues and cells (Portuguese Law 22/2007, June 29), with the approval of the Ethics Committee of the respective clinical institutions[36]. Three independent donors were used on the experiments (n=3, donor A: male donor, 35 years old; samples isolated in 2015, B: male donor, 73 years old; samples isolated in 2008, C: male donor, 38 years old; samples isolated in 2015). These cells belong to the cell bank available at Stem Cell Engineering Research Group (SCERG), iBB-Institute for Bioengineering and Biosciences at Instituto Superior Técnico (IST). Bone marrow samples were retrieved from Instituto Português de Oncologia Francisco Gentil, Lisboa-Portugal under collaboration with iBB-IST. Isolated cells were cryopreserved in liquid/vapor nitrogen tanks. Isolated human bone marrow MSCs (BM MSCs) were cultured on low-glucose Dulbecco’s Modified Eagle Medium (DMEM, Gibco Grand Island New York) supplemented with 10% fetal bovine serum (FBS MSC qualified, Gibco) and 1% antibiotic-antimycotic (Gibco) and kept at 37ºC, 5% CO2 and 21% O2 in a humidified atmosphere. The phenotype of MSCs under magnetic exposure was confirmed in our previous study[37]. Briefly, the cells were tested by flow cytometry, with and without magnetic exposure, for expression of cell surface markers indicative of MSCs, using a panel of mouse anti-human monoclonal antibodies (PE-conjugated) against: CD73+, CD90+, CD105+, CD14-, and human leukocyte antigen HLA-DR- (all from Biolegend). All cellular experiments with MSCs were performed between passages 4 and 7.
HUVECs used in this study are commercially available and were purchased from Lonza (Basel, Switzerland). HUVECs were expanded in commercial endothelial growth medium-2 (EGM-2, Lonza) and kept at 37ºC, 5% CO2 in a humidified atmosphere. Medium renewal was performed every 3-4 days. All cellular experiments with HUVECs were performed between passages 4 and 7.”
The point 2.4 should be adequately explained. Please expand this point appropriately. Did the authors use fluorescence typing? The authors must provide in detail all the data of their providers. This information is not specified in the materials and methods section.
The authors understand the concern of the Reviewer and consequently the Experimental section 2.4 was improved for a more detailed understanding of the method. MSC were labelled with AlamarBlue reagent for determination of cell proliferation, and an indication of cell viability, by confocal fluorescence microscopy. Briefly, AlamarBlue®cell viability reagent is an indigo-colored, non-toxic reagent able to detect metabolically active cells and is used for the quantitative analysis of cell viability and proliferation. These changes were included in page 6 lines 204.
“2.4. MSC proliferation values.
MSCs proliferative assessment was evaluated 24 h after magnetic exposure (static or dynamic) and compared with the initial density of 75,000 cell/cm3. MSCs were seeded on top of mGelatin scaffolds at a density of using expansion medium (DMEM + 10% FBS). As a control, MSCs were seeded as a monolayer on cell culture plates at a density of 75,000 cell/cm2. The metabolic activity of MSCs was evaluated using AlamarBlue® cell viability reagent (Molecular proves, Eugene, Oregon, U.S.), following manufacturer instructions. This non-toxic, cell-permeable reagent is a resazurin-based solution, blue in color and non-fluorescent, that functions as a cell health indicator by using the reducing power of living cells to quantitatively measure viability. MSCs treated with 10% (v/v) AlamarBlue® cell viability reagent were incubated at 37ºC in 5% CO2 chamber for 2 h. After entering the cells, the resazurin compound of AlamarBlue® is reduced to resorufin, a compound red in color and highly fluorescent. Fluorescence intensity of the supernatant of the cells was quantified in a range of 560-590 nm. Prior to analysis, a calibration curve for different human bone marrow MSC densities (10,000, 20,000, 50,000, 75,000, 100,000, 150,000 cells/mL) was used to convert the obtained metabolic values into cell numbers and determine the cell proliferation values associated with each group of conditions. All conditions were tested in triplicates during this experiment using three independent MSC donors (n=3).”
How do the authors justify the sample size for the publication to have a translational character?
The authors agree that the sample size may be a critical aspect and acknowledge the fact that having a sample size of n=3, added to the variability of MSCs depending on the donor, is not enough to justify clinical translation yet. However, the intention of the present study is to present a concept using in vitro experiments to show the future potential of this method in the clinical sector. The authors are aware of the need to optimize this method and to perform additional functional studies before planning the strategy for clinical translation of this method. The authors also intend to develop further studies which may provide information on the impact of the magnetic field at the molecular level, including the activation of pathways in the presence of the magnetic field. Some of these studies are presently ongoing. We also mentioned this issue over the last paragraphs of the Discussion section (page 16, line 624-627).
“Testing new donors, to decrease the effect of donor variability, and exploring the impact of the magnetic field over the secretion of other molecules by MSCs might contribute for the robustness of the study and the validation of the clinical applicability of this system.”
The description of the results has not improved. The figures are still not very explanatory, for example figure 4 is not suitable among others. Pay the figure 6 is not valuable. The reader cannot see anything. Magnification is not described. Please, the authors should enlarge it or split the figures. Authors must put readings on all panels. The authors must unify their figures and the way in which they present the results. This oil is critical and is currently not assessable. The data and p values must be specified at all times.
The authors followed the suggestions of the Reviewer and changed the Figures to the best of our perception of the Reviewers’ request. All the figures were enlarged as much as possible to allow for a better reading of the results. The captions of all the figures were improved with more detailed information, in particular Figures 4 and 5. The magnification scales included in different images were verified. The statistical analysis and p values were specified in every suitable panel. Captions of Figures 4 and 5 were modified as shown below:
“Figure 4. VEGF-A expression values using RT-PCR technique to compare the effect of static and dynamic modes (low and high frequency magnetic field) and magnetic intensities (0.08T and 0.45T) over MSCs secretome. Relative expression of VEGF-A from MSCs cultured on mGelatin scaffolds, in comparison with monolayer MSCs as a control condition, using three MSC donors (A, B, C) under independent conditions. Data presented as means + SD. Statistical significance was determined using two-way ANOVA; *p < 0.05 indicates a significant result; **p < 0.01 a very significant result, ***p < 0.001 a highly significant result, and ****p < 0.0001 an extremely significant result.”
“Figure 5. VEGF-A protein quantification for MSCs cultured on mGelatin and control condition (MSCs cultured on monolayer) for each experimental condition (static, lower and higher frequency magnetic field) and corresponding donor (A, B, C), according with the magnetic intensity (0.08T and 0.45T). Three MSC donors (A, B, C) were used under independent conditions. Data presented as means + SD. Statistical significance was determined using two-way ANOVA; *p < 0.05 indicates a significant result; **p < 0.01 a very significant result, ***p < 0.001 a highly significant result, and ****p < 0.0001 an extremely significant result.”
Figure 6. a) Representative images of HUVECs sprouting obtained from MSCs conditioned media cultured in each condition (mGelatin vs monolayer; 0.08T vs 0.45T; static vs lower frequency vs higher frequency magnetic field). Scale bar: 100 µm. b) Number of tubes and number of branch points measured for each analysis condition of the tube formation assay. c) Representative images of the tube formation controls and respective measurement of tubes number and branch points. Endothelial cell culture media, rich in growth factor supplements, was used as positive control. MSCs secretome, from MSCs cultured in polystyrene culture plate without magnetic exposure, was used as negative control. Scale bar: 100 µm. All data obtained from a pool of three independent MSC donors (A, B, C) and presented as means + SD. Statistical significance was determined using two-way ANOVA; *p < 0.05 indicates a significant result; **p < 0.01 a very significant result, ***p < 0.001 a highly significant result, and ****p < 0.0001 an extremely significant result.
The authors must specify the methodology of analysis and interpretation. How do they do and justify cell viability? Cycle? Why is flow cytometry not done specifically?
The authors understand that cell viability might be analyzed through different and more precise techniques than the ones used in this study, such as fixable LIVE/DEAD Kit for flow cytometry quantification of the viability of the cells. However, in the present work, we opted to use the LIVE/DEAD Kit for microscopic fluorescence analysis which also allows to distinguish the living and the dead cells by using specific probes to stain the cells with different coloring (green for living cells and red for dead cells). Nonetheless the Sub-section 2.6 in the Experimental section was improved in page 6 line 232 in the revised version of the manuscript to further describe the applied methodology, as indicated below.
“2.6. Cell viability and morphology assay.
MSCs were seeded on the scaffolds at a density of 75,000 cell/ cm3. To assess MSCs viability, cells were washed twice with PBS (Phosphate Buffer Saline, Dulbecco’s Sigma-Aldrich) after 24 h of magnetic application and stained for 20 min using LIVE/DEADTM Viability/Cytotoxicity Kit, containing calcein and ethidium homodimer probes, (ThermoFisher Scientific, L3224), according to the manufacturer protocol for mammalian cells. After washing with PBS, cell imaging was obtained using a Fluorescence microscope (Leica DM IL LED with EC3 camera system). The external membrane of living cells was stained with calcein probe and observed in a green coloring while dead cells were highlighted with a red nucleus by ethidium homodimer probe. The calcein staining of the membrane of the cells also allows for morphological analysis of the cells under the different experimental conditions.”
- The discussion is absolutely inappropriate. The authors have not remodeled anything. The discussion must have a translational character and have a more modern bibliography.
The authors understand the concern of the Reviewer, however as referred in the answer to comment 4, these results should be regarded as a proof of concept which successfully show the potential of the magnetic field combined with the used of magnetic-responsive scaffolds to regulate angiogenic effects (inducing either pro-angiogeneis and anti-angiogenesis). As also referred, the number of samples (3 donors) is yet too short for a clear discussion on the translational character of this methodology. For this reason, the authors agree to refer to potential clinic translation at the end of the Introduction and Discussion sections (page 3 line 98, page 16 line 622, respectively) and in Conclusion section (page 16, line 644) in the revised version of the manuscript. A deeper and clear discussion of this topic is dependent on the confirmation of these results through systematic analysis using samples from a higher number of donors couple with other studies.
A more modern bibliography would have also been very useful for us as a way to compare it with the methodology we used in this study. However, the use of magnetic field for biological applications is very underexplored at the moment and most of the existing studies that allow for proper discussion are not recent. Still, we considered them essential to refer to fundamental physical aspects of the magnetic field which may explain the observed magnetic field effect w. Nevertheless, we believe this further emphasizes the novel aspect of the developed work since there are no similar works in the literature at the moment.
We also tried to clarify the information in the discussion and highlight the clinical translation potential of the study more clearly. Major changes in the text can be found below (last paragraphs of the discussion and the conclusion).
“This report highlights the capacity to remotely regulate angiogenic behavior through the manipulation of magnetic dynamic regimes. The results evidenced the potential for future clinical translation with the development of novel magnetic-based therapeutic routes for several diseases. For instance, to use LF magnetic regime for anti-tumorigenic applications for cancer treatment, aiming to decrease tumor vascularization and delay cancer progression. Likewise, the condition using a mGelatin scaffold and static low-intensity magnetic field indicated enhancement of the angiogenic potential with possible applications to revascularize damaged tissues in several vascular diseases (e.g. vascular diseases, diabetes, atherosclerosis). Additionally, the potential impact of the magnetic stimulation on other molecules from the VEGF family, such as VEGF-C, might also offer perspectives on the application of this approach to promote lymphangiogenesis for the treatment of lymphatic conditions.
Despite these promising results, more exhaustive investigation would be required for a deeper and more complete understanding of the potential effect of magnetic field on other cell mechanisms involved in angiogenesis. Testing new donors, to decrease the effect of donor variability, and exploring the impact of the magnetic field over the secretion of other molecules by MSCs might also contribute to the robustness of the study and the validation of the clinical applicability of this system.”
“5. CONCLUSIONS
This study investigates the indirect impact of magnetic field on the angiogenic activity of HUVECs. The study was based on the ability to regulate VEGF-A expression by MSCs through changes in the intensity and dynamic regime of the applied magnetic field.
Lower magnetic field intensity (0.08 T), as well as high frequency magnetic variation regime, resulted in the increased production of growth factor VEGF-A comparatively to that observed at magnetic field with higher intensity (0.45 T) or applied under low frequency modes (LF modes), which translated into a positive angiogenic effect in HUVECs with the formation of a high number of tubes and branch points. Higher magnetic intensity of 0.45 T showed a significant decrease in the angiogenic potential of MSCs, consistent with the observed decreased cell number, suggesting increased mortality of MSCs when exposed to higher magnetic intensities. Lower frequency cyclic magnetic field regimes were also associated with an anti-angiogenic performance, demonstrated by the decreased secretion of VEGF-A in MSCs and by the lower number of tubes formed by HUVECs. Despite the better angiogenic response of MSCs in the presence of mGelatin, MSC monolayers (control) also demonstrated a satisfactory magnetic response, particularly evident under dynamic magnetic field regimes. From the clinical translational point of view, this work proves that the magnetic regulation of angiogenesis may be attained non-invasively in the presence and in some extent in the absence of magnetic support (magnetic-responsive gelatin scaffold) i.e. accounting with the magnetic susceptibility of MSCs. Overall, these results reveal that magnetic stimulation may either function as a tool to reduce tumor growth in association with cancer treatments (using LF operation modes) or as a possibility to promote the regeneration of diseased blood vessels (static and HF modes) in a contribution to vascular diseases treatments through cellular therapies. Although complementary studies are yet required for a deeper understanding of the cell mechanisms, the present investigation reveals a modulatory effect of magnetic field regimes on MSCs behavior, thus offering the possibility to regulate angiogenesis phenomena with potential contribute for future clinical applicability.”
Explain the supplementary material adequately. Authors must include all unmounted images as well (fluorescence, etc …).
The authors have confirmed the existence of readings in all panels, reformulated the captions to be more explanatory and improved the resolution of the images. Examples below.
Figure S1. Representative images of MSC distribution in the mGelatin scaffold under exposure to different magnetic field intensities (0.08 T, 0.45 T) and regimes (static, HF, LF). A LIVE/DEAD assay was performed and MSCs were stained with calcein (to identify living cells) and with ethidium bromide (to indicate dead cells). Scale bar: 200 µm. No significative cell patterning differences for MSCs exposed to static, LF or HF dynamic regimes were observed through comparative analysis of these images. However, a reduced MSCs density was observed upon exposure to LF conditions.
Figure S3. Representative images showing a randomly dispersed monolayer of HUVECs (brightfield image on the left, scale bar: 50 µm) and the identification of the endothelial lineage of the cells using CD31 (red color), specific marker of endothelial cells for immunofluorescence, and DAPI (blue color), to stain cell nuclei (immunofluorescence image on the right, scale bar: 50 µm).
Figure S4. Representative images of HUVECs stained with calcein (indicator of living cells) to show morphological differences between HUVECs cultured in monolayer, before cell maturation in the tube formation assay with a random cell distribution (left side, scale bar: 50 µm) and during tube formation functional experiment (right side, scale bar: 100 µm), where tube-like structures are formed. During tube formation, HUVECs were supplemented with the conditioned media from MSCs exposed to magnetic stimulation.
The authors must include a graphic summary.
The authors agree with the comment of the Reviewer and have added a graphic summary to the manuscript. The graphical abstract can be observed in the attached file.
Graphical abstract illustrating the study described within the manuscript. The graphical abstract illustrates the process of collecting MSCs from donors, culturing the cells on magnetic scaffolds and expose the cell culture to the remotely controlled magnetic system to provide cyclic variation of the magnetic field. The graphical abstract also shows the goal of the study to regulate angiogenesis in vitro. Proangiogenic and antiangiogenic effects are illustrated in the figure by supplementing a culture of HUVECs with conditioned media containing different concentrations of angiogenic genes obtained from the conditioned media of MSCs exposed to magnetic stimulation.
The conclusion is too long and the authors extend themselves without giving the reader a clear and precise idea of the results.
The authors thank the comment of the Reviewer and have decided to reduce the text of the Conclusion section. This text was also modified aiming to better clarify the impact of the study. The remodeled conclusion can be found below, as well as in the comment 7.
“5. CONCLUSIONS
This study investigates the indirect impact of magnetic field on the angiogenic activity of HUVECs. The study was based on the ability to regulate VEGF-A expression by MSCs through changes in the intensity and dynamic regime of the applied magnetic field.
Lower magnetic field intensity (0.08 T), as well as high frequency magnetic variation regime, resulted in the increased production of growth factor VEGF-A comparatively to that observed at magnetic field with higher intensity (0.45 T) or applied under low frequency modes (LF modes), which translated into a positive angiogenic effect in HUVECs with the formation of a high number of tubes and branch points. Higher magnetic intensity of 0.45 T showed a significant decrease in the angiogenic potential of MSCs, consistent with the observed decreased cell number, suggesting increased mortality of MSCs when exposed to higher magnetic intensities. Lower frequency cyclic magnetic field regimes were also associated with an anti-angiogenic performance, demonstrated by the decreased secretion of VEGF-A in MSCs and by the lower number of tubes formed by HUVECs. Despite the better angiogenic response of MSCs in the presence of mGelatin, MSC monolayers (control) also demonstrated a satisfactory magnetic response, particularly evident under dynamic magnetic field regimes. From the clinical translational point of view, this work proves that the magnetic regulation of angiogenesis may be attained non-invasively in the presence and in some extent in the absence of magnetic support (magnetic-responsive gelatin scaffold) i.e. accounting with the magnetic susceptibility of MSCs. Overall, these results reveal that magnetic stimulation may either function as a tool to reduce tumor growth in association with cancer treatments (using LF operation modes) or as a possibility to promote the regeneration of diseased blood vessels (static and HF modes) in a contribution to vascular diseases treatments through cellular therapies. Although complementary studies are yet required for a deeper understanding of the cell mechanisms, the present investigation reveals a modulatory effect of magnetic field regimes on MSCs behavior, thus offering the possibility to regulate angiogenesis phenomena with potential contribute for future clinical applicability.”
The authors have not checked the grammar of their manuscript. Authors should use the publisher's central services.
The authors took the suggestion of the Reviewer into consideration and have carefully and extensively revised and improved the grammar of the manuscript with the assistance of a linguistic expert. Linguistic corrections included in the revised version of the manuscript were marked in blue color.

Reviewer 4 Report
All the referres' comments have been satisfied and discussed.
Author Response
Magnetic field dynamic strategies for improved control of the angiogenic effect of mesenchymal stromal cells
Ana C. Manjua, Joaquim M.S. Cabral, Frederico Castelo Ferreira*, Carla A. M. Portugal*
Manuscript ID: polymers-1171347
Answers to the comments of Reviewer 4:
The authors would like to thank the Reviewer for the constructive feedback and useful comments and suggestions.
Comments of the Reviewer:
All the referres' comments have been satisfied and discussed.
